# JACKPOT: ALIGN ACTOR-POLICY DISTRIBUTION FOR SCALABLE AND STABLE RL FOR LLM

**Zhuoming Chen**[*]  **Hongyi Liu**[*]  **Yang Zhou**[*]  **Haizhong Zheng**  **Beidi Chen**
Carnegie Mellon University
{zhuominc,hongyil2,yangzho6,haizhonz,beidic}@andrew.cmu.edu

## ABSTRACT

Reinforcement learning (RL) has become an increasingly important paradigm for improving large language models (LLMs) on alignment, reasoning, and coding tasks, yet it remains extremely costly. The majority of training time is spent on rollouts. Allowing actor and policy distributions to differ could unlock substantial scalability and efficiency benefits, such as supporting large-batch or asynchronous training, and even enabling a lightweight rollout model. However, existing importance sampling–based corrections for distribution mismatch suffer from an inherent trade-off between stability and training performance. To tackle this problem, we propose Jackpot, which leverages Optimal Budget Rejection Sampling to directly reduce the gap between actor and policy distributions. For efficiency and stability in practical training, We introduce an efficient probability estimation strategy based on Top-$K$ logits with batch bias correction, and designs a stabilized Jackpot-PPO loss that jointly accounts for both the importance sampling ratio and the trust-region constraint in PPO. Empirically, our method achieves stable improvements in large-batch and asynchronous training, and in extreme off-policy training it substantially delays the onset of collapse and delivers competitive performance. Specifically, we achieve 20% improvement on AMC benchmarks and 8% AIME benchmarks over the off-policy baseline under $128\times$ actor-policy update ratio for Qwen3-4B-Base and $64\times$ for Qwen3-8B-Base, while achieving greater stability and better performance than prior off-policy RL methods under extreme settings. More details of the project are available at https://infini-ai-lab.github.io/jpt_website/.

## 1 INTRODUCTION

Reinforcement learning (RL) has demonstrated substantial effectiveness in the post-training of large language models (LLMs), yielding significant improvements in domains such as mathematics (Guo et al., 2025; Azerbayev et al., 2023), coding (Jimenez et al., 2023; Ouyang et al., 2025), and agentic tasks (Liu et al., 2023). Despite these successes, RL remains computationally expensive (Sheng et al., 2025; Fu et al., 2025; Zheng et al., 2025b), with the majority of the training cost, often exceeding $70\%$, attributed to rollouts, wherein LLMs generate solution trajectories for tasks in order to compute rewards. If actors and policies were allowed to follow different distributions, the scalability and efficiency of RL could be elevated to an entirely new level. For instance, such flexibility would make it possible to support large-batch or asynchronous training, thereby improving the utilization of serving systems (Zheng et al., 2025a). Moreover, quantized or sparse models, and even distilled smaller models, could be deployed as actors to further enhance efficiency. In practice, however, the mismatch between actor and policy distributions often leads to instability and severe degradation in performance (Liu et al., 2025), posing a fundamental barrier to the reliable adoption of these techniques.

When the distribution gap between the actor and the policy becomes large, existing importance sampling (IS)-based correction methods (Liu et al., 2025; Wu et al., 2025b; Fu et al., 2025) perform suboptimally compared with the baseline PPO. In practice, truncated importance sampling methods (TIS) either underperform or exhibit substantial convergence gap to the on-policy baseline when the truncation threshold is low or conservative, or TIS crashes before policy plateaus from RL training when the truncation threshold is set to a higher or aggressive value. The importance weight used by TIS is $\frac{p_{\text{ref}}(a)}{p_{\text{target}}(a)}$. Once the actor drifts too far, many tokens that the actor samples with high probability have very low probability under

---

[*]Equal contribution, alphabetically ordering based on lastnames

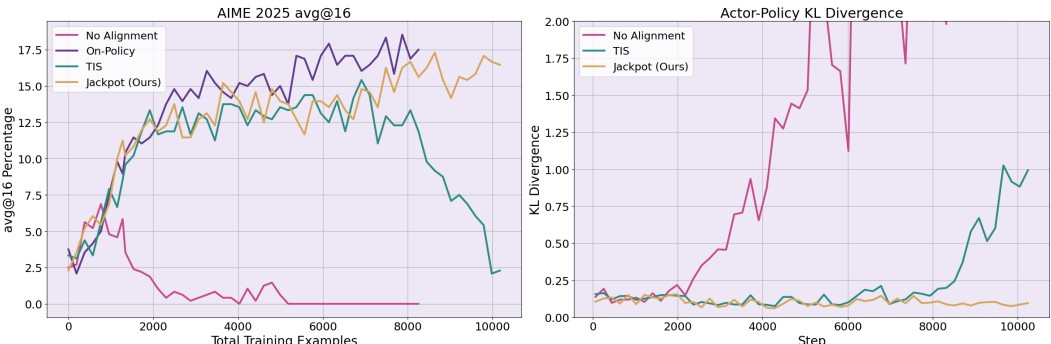

Figure 1: RL training requires actor-policy maintaining strong probability distribution alignment. When actor and policy aren't aligned, they will result in training collapse. Here we show training setting use a Qwen3-1.7B-Base model training rollout to train a Qwen3-8B-Base model policy. Without any alignment procedures, training collapses (pink). Prior method TIS (green) also show significant gap towards Qwen3-8B-Base on-policy baseline (purple), while collapsing, using TIS sees KL divergence also violently increasing. Our proposed method, Jackpot (yellow) maintains small KL divergence between actor and policy model probability distribution, while showing stable and competitive training convergence to on-policy setting.

the policy, since $p_{\text{inf}} > p_{\text{target}}$. These actor trajectories are effectively treated as low-likelihood samples by the policy, causing TIS to train on tokens the policy would never select at inference and creating a widening train-inference mismatch. This naturally raises the following question: *Can we directly modify the actor's sampling distribution and sampled trajectories to reduce its distributional gap to the policy?*

Rejection sampling, which can simulate a target distribution from an accessible proposal distribution, has a long history and has been widely applied in fields such as biology (Carrella et al., 2024), the social sciences, machine learning (Naesseth et al., 2016), and statistics (Martino & Míguez, 2011; Gilks et al., 1995). However, a direct application of rejection sampling is prohibitively expensive in the context of RL. Specifically, token $i$ must be accepted with probability $\frac{p_i}{\lambda q_i}$, where $\lambda = \max_i \frac{p_i}{q_i}$. For contemporary LLMs, which typically possess vocabularies exceeding 100,000 tokens, this constant $C$ can become extremely large, since the majority of token probabilities are exceedingly close to zero. As a result, nearly all tokens proposed by the actor are rejected, rendering naive rejection sampling impractical for large-scale RL and leading to prohibitively low sample efficiency. Fortunately, Optimal Budget Rejection Sampling (OBRS) (Verine et al., 2024) relaxes the requirement of classical rejection sampling and, although it does not enforce the actor distribution to be identical to the policy distribution, it provably reduces their distance and guarantees that for any rejection ratio the adjusted actor distribution is strictly closer to the policy distribution than the unadjusted one. This provides us with an opportunity fundamentally different from standard rejection sampling.

However, applying OBRS directly in RL systems introduces several technical challenges. First, PPO relies on the existence of a trust region to stabilize the training process, which means that modifying the actor probabilities through OBRS may compromise training stability. Second, in order to compute the true probabilities of the remaining tokens after the rejection process, OBRS requires access to the probabilities of all tokens in the vocabulary, which imposes significant memory overhead for modern LLMs with extremely large vocabularies.

In this paper, we propose JACKPOT, which consists of three key components. First, an OBRS-based masking mechanism ensures that the adjusted actor distribution remains strictly closer to the policy distribution. Second, an efficient probability estimation strategy is introduced, which leverages Top-$K$ logits together with batch-wise bias correction to approximate the full-vocabulary probabilities while mitigating memory overhead. Third, we design a stabilized JACKPOT-PPO loss that jointly accounts for both the importance sampling ratio and the trust region constraint in PPO, thereby preserving training stability.

To validate the effectiveness of our method, we consider two representative scenarios. **(1) Large-batch training.** In this setting, the LLM generates up to 128 mini-batches in a single rollout step, which enables more efficient utilization of serving system hardware resources. Empirically, we observe more than a $2\times$ improvement in end-to-end RL throughput compared to on-policy training. However, this comes at the cost of substantial policy drift during training, resulting in significant divergence between the rollout actor and the updated policy. **(2) Extreme off-policy training.** In this setting, we employ a fixed model for rollouts

that is different from the one being optimized. This configuration introduces a severe distributional mismatch, under which standard approaches typically fail and training collapses rapidly.

We organize the remainder of this paper as follows.

- In Section 2, we formally introduce the distribution mismatch between actors and policy, discuss its sources under different training scenarios, and review related work.
- In Section 3, we describe our application of OBRS to RL and validate its effectiveness through numerical experiments and empirical observations.
- In Section 4, we provide a detailed description of JACKPOT-PPO, including three key components: (i) OBRS masking, (ii) efficient probability estimation with Top-$K$ logits and batch bias correction, and (iii) a stabilized JACKPOT-PPO loss that jointly considers importance sampling ratios and PPO's trust-region constraint.
- In Section 5, we present experiments on Qwen models and mathematical reasoning tasks to validate JACKPOT. First, in large-batch training, our method maintains stable learning, outperforming offline and TIS baselines and approaching the performance of the online setting. Second, in extreme off-policy training, the proposed method substantially delays the onset of training collapse and achieves competitive performance.

Using JACKPOT, we achieve 20% improvement on AMC benchmarks and 8% AIME benchmarks over the off-policy baseline under 128× actor-policy update ratio for Qwen3-4B-Base and 64× for Qwen3-8B-Base, while achieving greater stability and better performance than prior off-policy RL methods under extreme settings. Overall, JACKPOT is simple to plug in, theoretically well-grounded, and holds the potential to enable more aggressive forms of off-policy RL.

## 2 BACKGROUND

In this section, we first formalize the distribution mismatch problem that arises in RL for LLMs. We then review several strands of related work of JACKPOT.

### 2.1 PROBLEM SETTING: PPO OBJECTIVE AND ACTOR-POLICY DISTRIBUTION MISMATCH

We begin with the clipped objective in PPO (Schulman et al., 2017), whose expectation can be written as

$$\mathcal{L}^{\text{PPO}}(\theta) = \mathbb{E}_{x \sim P_{\text{inf}}}\Big[\min\big(r_\theta(x)\hat{A}(x), \text{clip}(r_\theta(x), 1-\epsilon, 1+\epsilon)\hat{A}(x)\big)\Big] \tag{1}$$

where $r_\theta(x) = p_{\boldsymbol{\theta}_{\text{new}}}(x)/p_{\text{ref}}(x)$ is the likelihood ratio between the updated policy $p_{\boldsymbol{\theta}_{\text{new}}}$ and the reference policy $p_{\text{ref}}$, and $\hat{A}(x)$ denotes the estimated advantage at decision $x$. $p_{\text{inf}}$ is the inference distribution used to generate rollouts, $p_{\text{ref}}$ is the reference policy distribution assumed in the objective, and $p_{\boldsymbol{\theta}_{\text{new}}}$ is the updated policy distribution. In the standard process, it is assumed that $p_{\text{inf}} = p_{\text{ref}}$, but in practice this assumption is often violated, leading to actor–policy distribution mismatch.

Distribution mismatch is common and arises for several reasons, such as minor discrepancies between the inference engine and the reference policy by FSDP engines, the use of stale or asynchronous data, or rollouts generated by approximated models (e.g., quantized, sparsified, or distilled). Such mismatches can destabilize training and therefore require additional mechanisms to correct or mitigate their impact. .

### 2.2 RELATED WORK

**RL for LLM.** Reinforcement learning has been widely applied to LLMs to improve human alignment, reasoning, coding, and other complex tasks. Beyond PPO, memory efficient methods have been proposed, including ReMax (Li et al., 2023), RLOO (Ahmadian et al., 2024), and GRPO (Shao et al., 2024). In addition, methods such as SimPO (Meng et al., 2024) and DPO (Rafailov et al., 2023), which are based on offline RL, have also been employed for human alignment. RL training systems for LLMs, such as Verl (Sheng et al., 2025), AReal (Fu et al., 2025), TRL (von Werra et al., 2020), and OpenRLHF (Hu et al., 2024), have been developed to improve training throughput and scalability.

**Distribution Mismatch Correction in RL.** Actor-policy mismatch is a common problem that has long been studied, e.g. Impala Espeholt et al. (2018). To alleviate the actor-policy distribution gap, the method introduces a truncated importance sampling (TIS) to approximate the true PPO objective.

$$\mathcal{L}^{\text{PPO}}(\theta) = \mathbb{E}_{x \sim P_{\text{inf}}}\Big[\min\big(\tfrac{p_{\text{ref}}(x)}{p_{\text{inf}}(x)}, C\big)\min\big(r_\theta(x)\hat{A}(x), \text{clip}(r_\theta(x), 1-\epsilon, 1+\epsilon)\hat{A}(x)\big)\Big] \tag{2}$$

The truncation threshold $C$ is for maintaining the stability of the range of the importance ratio. Recently, several methods apply the truncated importance sampling method to RL of LLMs. Methods such as FlashRL (Liu et al., 2025), AReal (Fu et al., 2025), and LlamaRL (Wu et al., 2025b) address distribution mismatch by introducing (truncated) importance sampling ratios, typically of the form $p_{\text{ref}}/p_{\text{inf}}$, to correct the impact of mismatch on advantage estimation. From system perspective, FP32 LM heads (Liu et al., 2025) and deterministic LLM Inference (He & Lab, 2025) are implemented to mitigate the numerical issue of serving systems when rollout.

In this paper, we proposed JACKPOT. Our method is **orthogonal** to the above prior works. We directly modify $p_{\text{inf}}$ through rejection sampling and reweighting of the output probabilities so that the divergence between $p_{\text{inf}}$ and the target distributions $p_{\text{ref}}$ is provably reduced. Moreover, techniques such as TIS can be applied on top of this improved distribution to further correct the remaining mismatch in a complementary way. JACKPOT offers a mechanism that is shown to be effective in stabilizing RL training under severe mismatches.

## 3 CORRECTING DISTRIBUTION MISMATCH WITH BUDGETED REJECTION SAMPLING

One of the most fundamental challenges of modern RL framework for LLMs is the distributional mismatch between samples generated by our inference model (actor), or $p_{\text{inf}}$, and the true reference policy distribution, or $p_{\boldsymbol{\theta}_{\text{ref}}}$. One way is through Importance Sampling, or adding an importance ratio term $\frac{p_{\text{ref}}}{p_{\text{inf}}}$. However, as the trajectories are sampled and $p_{\text{inf}}(x)$ can be small, the importance ratio sometimes blows up in numeric value. In practice, $\min(\frac{p_{\text{ref}}}{p_{\text{inf}}}, C)$ is used Espeholt et al. (2018) to cap out the dangerously large values, leading to huge bias in correcting the distribution misalignment. **Instead of solely relying on Importance Sampling, can we modify $p_{\text{inf}}$ and the sampled trajectories directly so that it is closer in probability distribution to $p_{\text{ref}}$?**

One direct idea is Rejection Sampling (RS), or stochastically rejecting tokens in the tractories sampled with $p_{\text{inf}}$ based on the difference between the two distributions. Once a token is rejected, it contributes nothing to the loss and gradient calculation. While canonical rejection sampling could resolve this, its application here—using $p_{\text{inf}}$ as the proposal and $p_{\boldsymbol{\theta}_{\text{new}}}$ as the target—is impractical, as rejection sampling aims for identical probability distribution after correction. The potentially large divergence between these distributions would lead to a prohibitively low acceptance rate, essentially leading to most tokens being rejected. The data efficiency of RL training will be severely degraded, failing to meet practical requirements.

To overcome this, we adopt the principled approach of **Optimal Budgeted Rejection Sampling (OBRS)** (Verine et al., 2024). This technique reframes the problem: instead of demanding perfect adherence to the target distribution at the cost of sample efficiency, it seeks the optimal rejection rule that, for a given target acceptance rate (a "budget"), produces a distribution as close as possible to the target. This is precisely the trade-off our problem requires.

The method employs a scaled acceptance probability, where a scaling factor $\lambda$ is chosen to meet the desired sample throughput. A token $a$ sampled from the proposal $p_{\text{inf}}$ is accepted with probability $\alpha_C(a)$ defined as $\alpha_C(a) = \min\left(1, \frac{p_{\text{target}}(a)}{\lambda \cdot p_{\text{inf}}(a)}\right)$

### 3.1 NUMERICAL SIMULATION

Crucially, this calibration is highly efficient; the acceptance rate remains high even when there is a large initial KL divergence. The impact on distributional alignment is dramatic: a significant reduction in KL divergence is observed with high acceptance rates. By systematically damping the most extreme probability ratios, OBRS produces a distribution that is not only provably closer to the on-policy target but also primed to yield more stable and effective PPO/GRPO policy updates.

### 3.2 THEORETICAL GUARANTEES

OBRS possesses proven optimality. It has been established that for any desired average acceptance rate $\bar{\alpha} \in (0, 1]$, there exists a corresponding scaling factor $C$ that achieves it. Crucially, among all possible rejection rules that satisfy this budget, this scaled acceptance rule is the unique one that **minimizes the Kullback-Leibler (KL) divergence** to the target distribution $p_{\boldsymbol{\theta}_{\text{new}}}$. A formal statement and proof of this theorem are provided in Appendix §A.2. This guarantee ensures we are using the provably best method for trading sample efficiency for distributional accuracy.

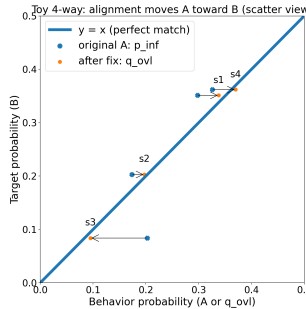 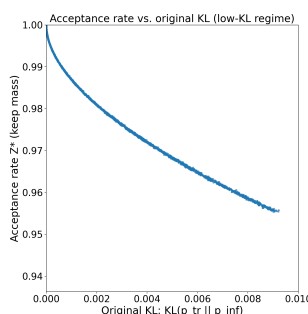 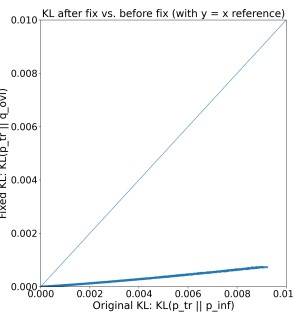

Figure 2: OBRS calibration results across three views: (a) per-token probability-ratio clipping pulls the model distribution toward the target, (b) acceptance remains high ($\approx 95\%$) even at large initial KL, and (c) overall KL is reduced by roughly an order of magnitude.

The scaling factor $C$ acts as an explicit control knob for this trade-off. A larger $C$ pushes the post-rejection distribution closer to the true target $p_{\theta_{\mathrm{new}}}$ at the expense of a lower acceptance rate, while a smaller $C$ boosts throughput at the cost of higher divergence. In our experiments, we find $C=1$ to be a robust default.

This formulation also guarantees that applying this sampling technique is always better than using the original inference distribution $p_{\mathrm{inf}}$ directly. The post-rejection distribution $p_{\mathrm{kept},C}$ is strictly closer to the target $p_{\theta_{\mathrm{new}}}$ in KL divergence than the original $p_{\mathrm{inf}}$ for any choice of $C>0$. We provide a summary of this proof tailored to our notation in Appendix §A. Our algorithm for implementing this procedure is also detailed in Appendix §A.

## 4 JACKPOT: DESIGN AND METHODOLOGY

In this section, we present details on the design considerations of JACKPOT. We show the token rejection criteria and reweighting procedures in Section, applying our rejection sampling to the PPO setup in Sections, and efficiency analysis in Section.

### 4.1 REJECTION AND REWEIGHTING

To bridge the inference probability distribution $p_{\mathrm{inf}}$ and the target distribution $p_{\mathrm{target}}$, we use the following critieria similar to Leviathan et al. (2023). For token sampled by $p_{\mathrm{inf}}$, x, we accept token with probability

$$\min(1, \frac{p_{\mathrm{target}}(x)}{\lambda p_{\mathrm{inf}}(x)}). \tag{3}$$

Note that the above equation is P(x accepted | x sampled by $p_{\mathrm{inf}}(x)$). Once the token x is rejected, it will be masked out and no longer participate in the loss and gradient propagation. After rejection, the distribution has expression:

$$P_{OBRS} = \frac{\min(p_{\mathrm{inf}}(x), \frac{p_{\mathrm{target}}(x)}{\lambda})}{\sum_{x'} \min(p_{\mathrm{inf}}(x'), \frac{p_{\mathrm{target}}(x')}{\lambda})}. \tag{4}$$

### 4.2 INTEGRATION WITH STANDARD PPO OBJECTIVE

Following Section 4.1 , we have the following PPO objective and applied Truncated Importance Sampling,

$$\mathcal{L}_{standard}^{\mathrm{PPO}}(\theta) = \mathbb{E}_{x \sim p_{\mathrm{inf}}} \Big[ f(x) \Big] = \mathbb{E}_{x \sim P_{\mathrm{inf}}} \Big[ \min\big( r_\theta(x) \hat{A}(x), \mathrm{clip}(r_\theta(x), 1-\epsilon, 1+\epsilon) \hat{A}(x) \big) \Big] \tag{5}$$

$$\mathcal{L}_{TIS}^{\mathrm{PPO}}(\theta) = \mathbb{E}_{x \sim P_{\mathrm{inf}}} \Big[ \min\big( \frac{p_{\mathrm{ref}}(x)}{p_{\mathrm{inf}}(x)}, C \big) f(x) \Big] \tag{6}$$

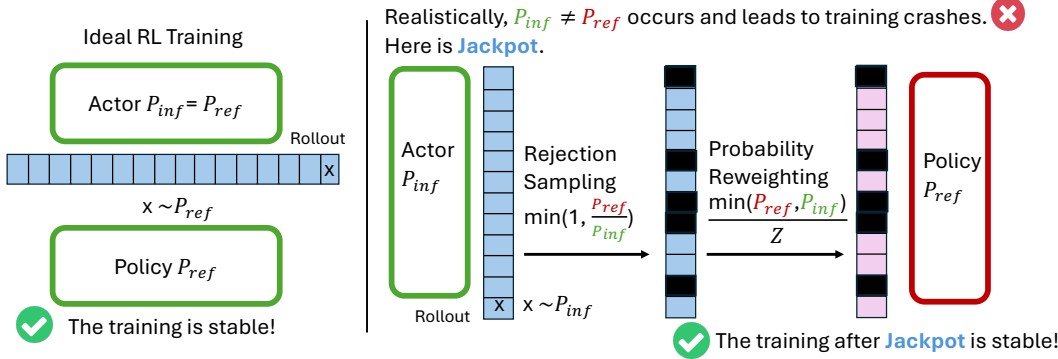

Figure 3: Illustration of JACKPOT Pipeline focusing on Optimal Budgeted Rejection Sampling (OBRS) and Reweighting Procedures

On top of TIS, we can further corrects the $p_{\text{inf}}$ of interest by using our rejection sampling critieria and reweighting by,

$$\min(\frac{p_{\text{ref}}(x)}{p_{\text{inf}}(x)},C)\rightarrow\min(\frac{p_{\text{ref}}(x)}{\frac{\min(p_{\text{ref}}(x')/\lambda,p_{\text{inf}}(x'))}{Z}},C)=\min(Z\cdot\max(\lambda,\frac{p_{\text{ref}}(x)}{p_{\text{inf}}(x)}),C) \tag{7}$$

where $Z$ is $\sum_{x'}\min(p_{\text{inf}}(x'),\frac{p_{\text{ref}}(x')}{\lambda})$. Therefore, instead of (6), we use the following PPO objective formulation,

$$\mathcal{L}^{\text{PPO}}_{ours}(\theta)=\mathbb{E}_{x\sim P_{\text{inf}}}\left[\min(Z\cdot\max(\lambda,\frac{p_{\text{ref}}(x)}{p_{\text{inf}}(x)}),C)\cdot f(x)\right] \tag{8}$$

### 4.3 WHICH POLICY TO APPROXIMATE?

Conventionally, we assume $p_{\text{inf}}=p_{\text{ref}}$, but empirically, we discover that for settings where RL training suffers from severe staleness, e.g. using large batch size or using asynchronized rollout/update cycles, approximating $p_{\text{inf}}\rightarrow p_{\text{new}}$ the latest updated policy edges out in performance. The rationale is that the reference policy is too stale and too distant to the latest updated policy to offer reliable trust region. In that case, we adjust the conventional PPO objective in (2) to the following from importance sampling,

$$\mathbb{E}_{x\sim P_{\text{ref}}}\left[f(x)\right]=\mathbb{E}_{x\sim P_{\text{new}}}\left[\frac{p_{\text{ref}}}{p_{\text{new}}}f(x)\right] \tag{9}$$

Then, we can approximate $p_{\text{new}}$ using $p_{\text{inf}}$ through our rejection sampling and reweighting. For the high staleness settings, we use the following approximating formulation,

$$\mathbb{E}_{x\sim p_{\text{new}}}\left[\frac{p_{\text{ref}}}{p_{\text{new}}}f(x)\right]\leftarrow\mathbb{E}_{x\sim p_{\text{inf}}}\left[\min(\frac{p_{\text{new}}}{p^*_{\text{inf}}},C_1)\cdot\min(\frac{p_{\text{ref}}}{p_{\text{new}}},C_2)\cdot f(x)\right] \tag{10}$$

where $p^*_{\text{inf}}$ is the corrected distribution through rejection sampling and reweighting.

We then have the JACKPOT objective,

$$\mathcal{L}^{\text{PPO}}_{ours}(\theta)=\left[\min(Z\cdot\max(\lambda,\frac{p_{\text{new}}(x)}{p_{\text{inf}}(x)}),C_1)\cdot\min(\frac{p_{\text{ref}}}{p_{\text{new}}},C_2)\cdot f(x)\right] \tag{11}$$

Throughout our experiments, we use $\lambda=1$. **We offer the user to either use (8) to align to $p_{\text{ref}}$ or $p_{\text{new}}$ depending on their target policy desired under their use cases.**

### 4.4 STABILIZATION AND FEASIBILITY CHALLENGES

Implementing JACKPOT directly faces a huge challenge of computational feasibility. Note that the weight's normalization constant, $Z$, requires a sum over the entire vocabulary ($|\mathcal{V}|>100,000$), creating a crippling memory bottleneck from storing full logit vectors (`batch_size × seq_len × vocab_size`). This severely restricts batch sizes, directly undermining the efficiency OBRS is intended to provide. Therefore, transforming this principled approach into a production-ready algorithm requires non-trivial engineering: we must introduce mechanisms to both bound the importance weights for stability and develop a computationally efficient, low-bias estimator for the normalization constant. To overcome the computational bottleneck of calculating $Z$, we employ a top-k approximation, which we then de-bias empirically.

### 4.4.1 TOP-K APPROXIMATION

The probability mass of language models is typically concentrated in a small subset of the vocabulary. We leverage this property by approximating the sum over $\mathcal{V}$ with a sum over a much smaller set, $\mathcal{V}_k$, which contains the most likely tokens from both the inference and current policies. Specifically, let top-k$(p)$ be the set of $k$ tokens with the highest probability under distribution $p$. We define our approximation set as the union:

$$\mathcal{V}_k = \text{top-k}(p_{\text{inf}}) \cup \text{top-k}(p_{\boldsymbol{\theta}_{\text{new}}})$$

The union is crucial because a token might be highly probable under one distribution but not the other, and the $\min$ function makes these overlapping regions important. The approximate normalization constant, $Z_{\text{approx}}$, is then: $Z_{\text{approx}} = \sum_{a' \in \mathcal{V}_k} \min\left(p_{\text{inf}}(a'), \frac{p_{\text{target}}(a')}{\lambda}\right)$

### 4.4.2 BIAS CORRECTION

While efficient, this top-k approximation introduces a systematic bias. Since the terms in the sum are non-negative, omitting tokens from the full vocabulary $\mathcal{V}$ can only decrease the total sum. Therefore, our approximation is a consistent underestimation of the true value:

$$\mathbb{E}[Z_{\text{approx}}] \leq Z$$

For k=20, . This bias could systematically alter the scale of the gradients during training. Fortunately, there is an elegant way to correct this. A key property of the framework is that the true normalization constant $Z$ is exactly equal to the expected acceptance rate, $\bar{\alpha}$:

$$\bar{\alpha} = \sum_{a \in \mathcal{V}} p_{\text{inf}}(a) \cdot \min\left(1, \frac{p_{\text{target}}(a)}{\lambda \cdot p_{\text{inf}}(a)}\right) = \sum_{a \in \mathcal{V}} \min\left(p_{\text{inf}}(a), \frac{p_{\text{target}}(a)}{\lambda}\right) = Z.$$

During the data collection phase (Algorithm 1, Phase 1), we can compute an unbiased empirical estimate of $\bar{\alpha}$ from the observed samples:

$$\hat{\bar{\alpha}} = \frac{\text{Number of accepted samples}}{\text{Total number of proposed samples}}$$

This gives us two estimators for $Z$: the low-variance but biased $Z_{\text{approx}}$, and the unbiased but higher-variance $\hat{\bar{\alpha}}$. We can combine them to create a de-biased, low-variance estimator. We compute a batch-wide calibration factor, $\kappa$, by dividing the empirical acceptance rate by the batch-averaged $Z_{\text{approx}}$:

$$\kappa = \frac{\hat{\bar{\alpha}}}{\frac{1}{B} \sum_{i=1}^{B} Z_{\text{approx}}^{(i)}}$$

where $B$ is the number of samples in the batch. We then apply this scalar correction to each per-token $Z_{\text{approx}}$ value used in the loss calculation. This procedure scales our efficient top-k estimate to match the true expected value observed in practice, effectively removing the bias while retaining the computational benefits and lower variance of the top-k approach.

## 4.5 IMPLEMENTATION OVERHEAD ANALYSIS

JACKPOT is lightweight. First, JACKPOT requires no additional trajectories sampled, as all the experiments we conducted in the extensive empirical studies section are using the same rollout width as the on-policy baseline. **A critical distinction to Leviathan et al. (2023) is that JACKPOT doesn't reject all tokens from the first place in the trajectory where a token first rejects and resample from where. Instead, JACKPOT only mask out tokens using our rejection critieria, and no additional trajectories sampling needed.** Second, no additional `log_prob` computation needed, since both $p_{\text{ref}}$ and $p_{\text{new}}$ will be computed by the standard objective (5), and JACKPOT only needs to reuse these probability distributions already computed. Third, no modification required on vLLM. Since JACKPOT doesn't required special operator or numeric precisions, we directly based our implementation on standard vLLM for rollout, without relying on custom kernels in vLLM.

JACKPOT indeed add minor additional overhead to standard PPO objective computation. The added computation comes from forcing vLLM to return `top-K` logprobs. Fortunately, since we only use k=20 for our runs, the added extra compute only added less than 3% to the total compute. In contrast, JACKPOT helps models be trained using $64\times$ or higher batch sizes by drastically alleviating the lack of convergence from the staleness of the actor model. Thus, comparing with the small batch and on-policy performance, we achieve more than 4 times speedup.

---

**Algorithm 1** The Jackpot Algorithm

---

**Require:** Policies: current $p_{\text{new}}$, reference $p_{\text{ref}}$, inference $p_{\text{inf}}$.
**Require:** Hyperparameters: OBRS threshold $\lambda$, PPO clip $\epsilon$, Jackpot clips $c_1, c_2$, top-$k$ count.
 1: **Convention:** SG($\cdot$) denotes the stop-gradient operation.
 2: **Implementation note:** Jackpot only reweights quantities from the *standard* rollout and PPO/GRPO forward passes; it does *not* perform extra model forward passes or trajectory recomputation.
 3: **Phase 1: Efficient Rollout (Standard Generation)**
 4: Initialize experience buffer $\mathcal{D} \leftarrow \emptyset$.
 5: **for** each trajectory sampling step $t$ **do**
 6:     Single forward pass of $p_{\text{inf}}(\cdot \mid s_t)$, sample $a_t \sim p_{\text{inf}}(\cdot \mid s_t)$.
 7:     From the same forward, compute and store top-$k$ log-probabilities of $p_{\text{inf}}$: $\text{TopK}_{\text{inf}}(s_t)$.
 8:     Store $(s_t, a_t, p_{\text{inf}}(a_t \mid s_t), \text{TopK}_{\text{inf}}(s_t))$ (plus rewards, values, etc.) in buffer $\mathcal{D}$.
 9: **end for**
10: Compute advantages $\hat{A}_t$ using collected trajectories.
11: **Phase 2: PPO Update with Jackpot Reweighting**
12: **for** each mini-batch sampled from $\mathcal{D}$ **do**
13:     **// 1. Standard PPO Computation (reused by Jackpot)**
14:     Forward pass $p_{\text{new}}$ and $p_{\text{ref}}$ on the mini-batch to get logits, $p_{\text{new}}(a_t \mid s_t)$, $p_{\text{ref}}(a_t \mid s_t)$, and $\text{TopK}_{\text{new}}(s_t)$.
15:     Compute policy ratio: $r_t(\boldsymbol{\theta}) = \dfrac{p_{\text{new}}(a_t \mid s_t)}{p_{\text{ref}}(a_t \mid s_t)}$.
16:     Compute vanilla PPO objective: $\mathcal{L}_{\text{PPO}} = \min\big(r_t(\boldsymbol{\theta})\hat{A}_t, \text{clip}(r_t(\boldsymbol{\theta}), 1-\epsilon, 1+\epsilon)\hat{A}_t\big)$.
17:     **// 2. Efficient $Z$-Approximation and Bias Correction (no extra forward passes)**
18:     Construct approximation set $\mathcal{V}_k = \text{TopK}_{\text{inf}}(s_t) \cup \text{TopK}_{\text{new}}(s_t)$.
19:     Compute
$$Z_{\text{approx}} = \sum_{x \in \mathcal{V}_k} \min\left(p_{\text{inf}}(x \mid s_t), \frac{p_{\text{new}}(x \mid s_t)}{\lambda}\right).$$
20:     Estimate correction factor $\kappa$ using the OBRS-based bias-correction procedure described in Sec. 4.4.2 (e.g., from batch-level OBRS statistics).
21:     Set corrected normalizer $Z_t \leftarrow \kappa \cdot Z_{\text{approx}}$.
22:     **// 3. Jackpot Weight Calculation**
23:     OBRS weight: $w_{\text{OBRS}} = Z_t \cdot \max\left(\lambda, \frac{p_{\text{new}}(a_t \mid s_t)}{p_{\text{inf}}(a_t \mid s_t)}\right)$.
24:     $\rho_{\text{jackpot}} = \min(w_{\text{OBRS}}, c_1) \cdot \min\left(\frac{p_{\text{ref}}(a_t \mid s_t)}{p_{\text{new}}(a_t \mid s_t)}, c_2\right)$.
25:     **// 4. Apply Weight to Loss**
26:     $\mathcal{L}_{\text{final}} = \text{SG}(\rho_{\text{jackpot}}) \cdot \mathcal{L}_{\text{PPO}}$.
27:     Update policy parameters new using gradient of $-\mathcal{L}_{\text{final}}$.
28: **end for**

---

## 5 EMPIRICAL VALIDATION

In this section, we comprehensively test our method on two different and challenging misalignment settings. First, we evaluate our method on an extremely large inference batch size, while keeping the training mini-batch size the same. At the end of each training-inference cycle, the staleness of the model weights in the inference server will be significantly amplified. Secondly, we evaluate our method using two separate models for training and rollout, an extreme setting where the output distribution gap is more severe than usual staleness in off-policy RL settings. We show that our technique enables the training model to better benefit from tokens from a completely distinct model. Because of the limit in space, we list detailed ablation studies on threshold selection and `top-K` analysis in the Appendix C.

### 5.1 INFERENCE WITH LARGE BATCH SIZE AND TRAINING WITH MUCH SMALLER BATCH SIZE

RL training using a much larger batch size for generation and a smaller batch size for training is common in practice. It is usually due to either system limitation, where the generation is highly parallelizable and relatively memory-light compared to training (saving optimizer states, etc.), or it is for better stability as

Table 1: Evaluation scores across benchmarks. TIS + Adjustment is explained in Section B.1.

| Models / Methods | GSM8K Mean@4 | MATH-500 Mean@4 | AMC22 & 23 Mean@4 | AMC12 2024 Mean@4 | AIME24 Mean@16 | AIME24 Pass@16 | AIME25 Mean@16 | AIME25 Pass@16 |
|---|---|---|---|---|---|---|---|---|
| **Qwen3-4B-Base on DeepScaleR-Preview Dataset** (rollout batch size = 2048; train batch size = 32; 64×) | | | | | | | | |
| On Policy | 92.19 | 81.55 | 58.43 | 51.11 | 23.12 | 33.13 | 22.91 | 30.95 |
| Off Policy | 88.04 | 71.15 | 39.15 | 29.44 | 13.96 | 23.03 | 11.04 | 18.61 |
| TIS | 89.67 | 72.00 | 42.77 | 31.67 | 11.88 | 17.56 | 11.25 | 17.28 |
| TIS + Adjustment | **92.76** | 79.50 | **57.22** | 43.33 | 18.75 | 26.03 | 17.71 | 24.61 |
| Jackpot (Ours) | 92.24 | **80.05** | 53.92 | 50.00 | **20.63** | **29.48** | **18.13** | **23.63** |
| **Qwen3-4B-Base on DeepScaleR-Preview Dataset** (rollout batch size = 4096; train batch size = 32; 128×) | | | | | | | | |
| Off Policy | 79.70 | 60.20 | 33.00 | 24.44 | 8.00 | 15.73 | 5.00 | 11.00 |
| TIS + Adjustment | 20.70 | 19.10 | 7.80 | 5.00 | 1.00 | 4.00 | 1.00 | 2.00 |
| Jackpot (Ours) | **92.00** | **80.00** | **51.20** | **47.22** | **19.16** | **24.58** | **18.52** | **25.08** |
| **Qwen3-8B-Base on DeepScaleR-Preview Dataset** (rollout batch size = 2048; train batch size = 32; 64×) | | | | | | | | |
| On Policy | 94.24 | 93.99 | 28.95 | 54.44 | 28.95 | 37.89 | 22.50 | 28.54 |
| Off Policy | 91.05 | 77.15 | 50.60 | 40.00 | 18.54 | 28.67 | 14.16 | 21.98 |
| TIS + Adjustment | 93.85 | 82.55 | 60.54 | 48.33 | 24.58 | 35.06 | 20.00 | 22.90 |
| Jackpot (Ours) | **94.01** | **83.05** | **63.55** | 54.44 | 26.87 | 36.23 | 20.41 | 26.57 |

Table 2: AMC22&23 results of two model training using various methods across the training steps. Mean@k/Pass@k.

| AMC22 & 23 | Rollout Model | Train Model | 0 | 20 | 30 | 40 | 50 | 60 | 70 |
|---|---|---|---|---|---|---|---|---|---|
| Vanilla GRPO | Q2.5-1.5B-IT | Qwen2.5-3B-Base | 18.1/48.2 | 31.8/59.0 | 32.2/61.5 | 33.3/59.0 | 30.0/49.4 | 22.4/42.3 | 5.7/16.9 |
| TIS | Qwen2.5-1.5B-IT | Q2.5-3B-Base | 18.1/48.2 | 28.9/57.8 | 28.2/54.2 | 32.5/68.7 | 26.8/56.6 | 0/0 | 0/0 |
| Jackpot (ours) | Q2.5-1.5B-IT | Qwen2.5-3B-Base | 18.1/48.2 | 29.4/59.0 | 31.2/57.8 | 33.9/60.2 | 31.9/62.7 | 28.8/54.2 | 13.4/31.3 |
| AMC22 & 23 | Rollout Model | Train Model | 0 | 20 | 30 | 40 | 50 | 60 | 70 |
| Vanilla GRPO | Q2.5-MATH-1.5B-IT | Qwen2.5-3B-Base | 18.1/48.2 | 23.3/60.2 | 22.6/57.8 | 24.7/53.0 | 17.0/43.4 | 3.9/14.5 | 0 |
| TIS | Q2.5-MATH-1.5B-IT | Qwen2.5-3B-Base | 18.1/48.2 | 25.8/59 | 25.0/59.0 | 21.7/55.4 | 0 | 0 | 0 |
| Jackpot (ours) | Q2.5-MATH-1.5B-IT | Qwen2.5-3B-Base | 18.1/48.2 | 27.3/57.8 | 29.4/63.9 | 26.7/59.0 | 28.6/60.2 | 22.4/54.2 | 2.9/16.9 |

advocated by Schulman et al. (2017). However, the delay in updates results in staleness, which we magnify for evaluating our method.

We use the best results under 30k examples for Qwen3-4B-Base as our metrics, and the best under 50k for Qwen3-8B-Base. [1]. The results of our run are summarized in table 1. On-policy RL generally converges much faster than off-policy runs. We observed that for less aggressive off-policy settings, our method's margin is quite small. However, for extreme settings, such as 4096-32 for Qwen3-4B-Base and 2048-32 for Qwen3-8B-Base. We observed consistently that TIS-adjusted crashes occurred way earlier than our settings, often resulting in inferior results. For example, Qwen3-4B-Base TIS-adjusted fails to converge from the beginning. Also, as shown in Figure fig. 4 (a), TIS-adjusted also crashes way earlier than ours under the 2048-32 setting. Across all settings we tested, our method greatly improves the speed of convergence of the off-policy RL, while also resulting in performance numbers comparable to the on-policy runs.

Besides, as a much less aggressive setting, KV FP8 quantization can result in a performance crash as shown in Figure 4(b). Here, we apply only our rejection sampling algorithm, without clipping/truncation or any similar tricks. We show our performance recovers from crashes.

## 5.2 EXTREME OFFPOLICY SETTINGS

In this section, we demonstrate our method's effectiveness on the extreme setting where the model under inference and training are fundamentally two separate models. Specifically, we hypothesize that our sampling algorithm can always filter out useful tokens for the model training, even given a completely exotic output response. We use Qwen2.5-3B-Base as the trainer model, and we adopt different inference models from Qwen2.5-1.5B-Instruct to Qwen2.5-Math-1.5B-Instruct, trained on MATH-8K Hendrycks et al. (2021) dataset. We find out that our method can still improve itself under such extreme settings.

---

[1]Qwen3-4B-Base on-policy run crashes at 30k examples.

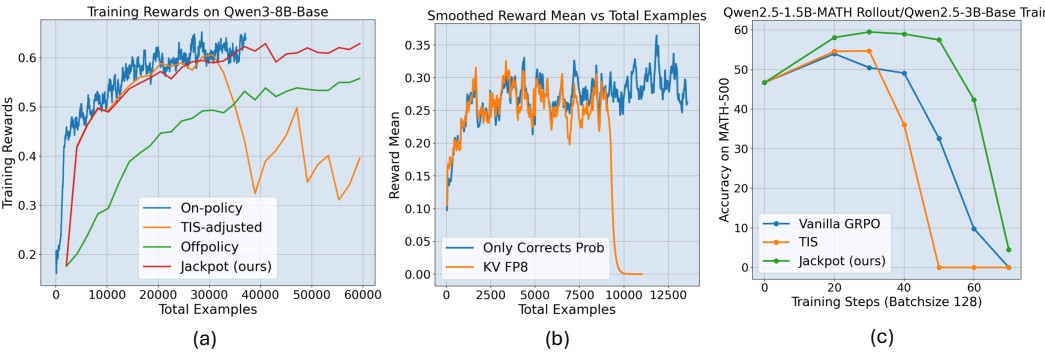

Figure 4: Overview of our empirical experiment results. (a) shows our proposed method's strength in correcting stale inference model distribution, and approaches the on-policy trend while maintaining training stability; (b) shows our proposed method to align output distribution can correct RL training instability even if applied alone; (c) The strength in our model's ability to align two different distributions allows us to put it in the most extremely misalignment setting where the rollout and the training model are separate of different architecture, our method shows early glimpse of hope and beats the baseline.

Shown in Figure 4 (c), we see that our run green curve shows clear improvement over our baseline GRPO and TIS settings. It even increases by 12% on MATH-500 under such an extreme misaligned setting. More results are summarized in Table 2, showing that the improvement also exists on AMC problems. We believe our method offers a new possibility to unlock much more scalability in RL efficiency and performance.

## 6 CONCLUSION

We propose Jackpot, which leverages Optimal Budget Rejection Sampling to directly reduce the gap between actor and policy distribution. Empirically, our method achieves stable improvements in large-batch and asynchronous training and also demonstrates stability under extreme misalignment settings.

## ACKNOWLEDGEMENT

We would like to thank computing resources of NVIDIA. This work was partially supported by Google Research Award, Google ML & System Junior Faculty Award, Amazon Research Award, Fireworks AI, Intel, Li Auto, Moffett AI, and CMU CyLab Seed funding. This material is also based upon work supported by the National Science Foundation under Grant No. 2504353 and IARPA. Any opinions, findings, and conclusions or recommendations expressed are those of the authors and do not necessarily reflect the views of the National Science Foundation.

ETHICS STATEMENT

Our study is centered on the development of reinforcement learning techniques for large language models. The research does not involve the use of human subjects, personal data, or other sensitive information. All datasets employed are openly accessible and commonly utilized within the research community. We recognize that advancements in LLMs may lead to societal risks, such as the potential misuse for generating harmful or deceptive outputs. To address such concerns, this work is conducted strictly within controlled academic environments, with the primary objective of enhancing the robustness and efficiency of training methodologies.

REPRODUCIBILITY STATEMENT

We have provided a comprehensive account of implementation details, hyperparameters, and experimental configurations in both the main text and the appendix. This documentation is intended to ensure that other researchers are able to independently reproduce our findings without ambiguity.

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

## A ANALYSIS OF OBRS

This appendix provides the theoretical foundation for OBRS. We first formally define the post-rejection distribution that results from our method. We then prove two key results:

1. **Optimality:** For any desired sample efficiency (i.e., acceptance rate), OBRS is the unique optimal rejection mechanism that produces a post-rejection distribution closest to the target $p_{\theta_{\text{new}}}$ in terms of KL divergence (Theorem 1).

2. **Monotonic Improvement:** The post-rejection distribution monotonically approaches the target distribution as the scaling factor $C$ increases. This guarantees that for any $C > 0$, OBRS reduces the KL divergence compared to using the proposal $p_{\text{inf}}$ directly. (Proposition 1).

For notational clarity, we consider the distributions over tokens for a single, fixed prompt and omit the explicit conditioning. Let $p_t \equiv p_{\theta_{\text{new}}}$ denote the target distribution and $p_p \equiv p_{\text{inf}}$ denote the proposal distribution.

---

**Algorithm 2** Implementation of Optimal Budgeted Rejection Sampling

---

**Require:** Proposal distribution $p_{\text{inf}}$, Target distribution $p_{\theta_{\text{new}}}$
**Require:** Scaling factor $C > 0$, Number of samples to accept $N$
**Ensure:** Set of accepted samples $S_{\text{kept}}$
 1: Initialize $S_{\text{kept}} \leftarrow \emptyset$
 2: **while** $|S_{\text{kept}}| < N$ **do**
 3:     Sample a token $a \sim p_{\text{inf}}(\cdot)$
 4:     Calculate acceptance probability $\alpha \leftarrow \min\left(1, \frac{p_{\theta_{\text{new}}}(a)}{C \cdot p_{\text{inf}}(a)}\right)$
 5:     **if** $U(0,1) < \alpha$ **then**
 6:         Add $a$ to $S_{\text{kept}}$
 7:     **end if**
 8: **end while**
 9:
10: **return** $S_{\text{kept}}$

---

### A.1 THE POST-REJECTION DISTRIBUTION

Recall from Definition 1 that OBRS accepts a token $a \sim p_p(a)$ with probability $\alpha_C(a) = \min\left(1, \frac{p_t(a)}{C \cdot p_p(a)}\right)$. The unnormalized probability of sampling and keeping a token $a$ is $p_p(a) \cdot \alpha_C(a)$, which simplifies to $\min\{p_p(a), p_t(a)/C\}$.

The overall probability of accepting any token, which we denote as the acceptance rate $Z_C$, is the sum over all possible tokens:

$$Z_C = \sum_{a \in \mathcal{A}} \min\left\{p_p(a), \frac{p_t(a)}{C}\right\}.$$

The distribution of the tokens that are kept, which we call the post-rejection distribution $p_{\text{kept},C}$, is therefore:

$$p_{\text{kept},C}(a) = \frac{\min\{p_p(a), p_t(a)/C\}}{Z_C}.$$

Special cases clarify the role of $C$: as $C \to 0$, $Z_C \to 1$ and $p_{\text{kept},C} \to p_p$ (all tokens are kept). As $C \to \infty$, $Z_C \to 0$ and $p_{\text{kept},C} \to p_t$ (perfect alignment with vanishing throughput). Standard rejection sampling is the special case where $C \geq \max_a(p_t(a)/p_p(a))$.

## A.2 Optimality for a Fixed Acceptance Budget

We first establish that OBRS is not merely a heuristic but is the provably optimal strategy for minimizing distributional error given a fixed efficiency budget.

**Theorem 1** (Budgeted Optimal Acceptance). *Fix a target acceptance rate (budget) $z \in (0, 1]$. Among all possible token-wise acceptance rules $\alpha : \mathcal{A} \to [0, 1]$ that satisfy the budget constraint $\mathbb{E}_{a \sim p_p}[\alpha(a)] = \sum_a p_p(a)\alpha(a) = z$, the rule that generates a post-rejection distribution $p_{kept}(a) \propto p_p(a)\alpha(a)$ minimizing the Kullback-Leibler (KL) divergence $KL(p_t \| p_{kept})$ is uniquely given by the OBRS rule:*

$$\alpha^\star(a) = \min\left(1, \frac{p_t(a)}{\lambda \cdot p_p(a)}\right)$$

*for some constant $\lambda > 0$ (equivalent to $C$) whose value is determined by the budget $z$.*

*Proof.* The post-rejection distribution is $p_{\text{kept}}(a) = p_p(a)\alpha(a)/z$. The KL divergence is:

$$\begin{aligned}
\mathrm{KL}(p_t \| p_{\text{kept}}) &= \sum_a p_t(a) \log \frac{p_t(a)}{p_{\text{kept}}(a)} \\
&= \sum_a p_t(a) \log \frac{p_t(a)}{p_p(a)\alpha(a)/z} \\
&= \underbrace{\sum_a p_t(a) \log \frac{p_t(a)}{p_p(a)} + \log z}_{\text{constant w.r.t. } \alpha} - \sum_a p_t(a) \log \alpha(a).
\end{aligned}$$

Minimizing $\mathrm{KL}(p_t \| p_{\text{kept}})$ is therefore equivalent to maximizing $\sum_a p_t(a) \log \alpha(a)$ subject to the constraints:

1. $\sum_a p_p(a)\alpha(a) = z$ (budget constraint)

2. $0 \leq \alpha(a) \leq 1$ for all $a \in \mathcal{A}$ (valid probability constraint)

This is a convex optimization problem. The Lagrangian is:

$$\mathcal{L} = -\sum_a p_t(a) \log \alpha(a) + \lambda \left(\sum_a p_p(a)\alpha(a) - z\right) + \sum_a \mu_a(\alpha(a) - 1) - \sum_a \nu_a \alpha(a)$$

where $\lambda, \mu_a, \nu_a$ are the KKT multipliers. From the stationarity condition $\frac{\partial \mathcal{L}}{\partial \alpha(a)} = 0$, we get $-\frac{p_t(a)}{\alpha(a)} + \lambda p_p(a) + \mu_a - \nu_a = 0$. The complementary slackness conditions imply that if $0 < \alpha(a) < 1$, then $\mu_a = \nu_a = 0$, which gives $\alpha(a) = \frac{p_t(a)}{\lambda p_p(a)}$. If $\alpha(a) = 1$, then $\nu_a = 0$, which requires $\lambda p_p(a) \geq p_t(a)$. If $\alpha(a) = 0$, this form is not well-defined, but the logic holds. Combining these cases, the optimal rule is to cap the acceptance ratio at 1:

$$\alpha^\star(a) = \min\left(1, \frac{p_t(a)}{\lambda p_p(a)}\right).$$

The Lagrange multiplier $\lambda$ is chosen to meet the budget constraint $\sum_a p_p(a)\alpha^\star(a) = z$. Uniqueness follows from the strict concavity of the log objective function. $\square$

## A.3 Guaranteed KL Divergence Reduction

Next, we show that our method provides a guaranteed improvement over the proposal distribution $p_p$ and that this improvement is monotonic in the control parameter $C$.

**Proposition 1** (Monotonic KL Contraction). *The function $G(C) = KL(p_t \| p_{kept,C})$ is non-increasing for $C \in (0, \infty)$. It is strictly decreasing wherever the set of tokens $\{a \mid p_t(a) < C \cdot p_p(a)\}$ has non-zero probability mass under $p_t$.*

*Proof.* Let $\rho(a) = p_t(a)/p_p(a)$. We partition the vocabulary $\mathcal{A}$ into two sets: $A_C = \{a \mid \rho(a) > C\}$ and $B_C = \{a \mid \rho(a) \le C\}$. On any open interval of $C$ where this partition is constant, we can write the KL divergence $G(C) = -\sum_a p_t(a)\log p_{\mathrm{kept},C}(a)$ as:

$$G(C) = -\sum_{a \in A_C} p_t(a)\log\frac{p_p(a)}{Z_C} - \sum_{a \in B_C} p_t(a)\log\frac{p_t(a)/C}{Z_C}$$

Differentiating with respect to $C$ (and noting that only $Z_C$ depends on $C$), we get $\frac{dG}{dC} = -\frac{d}{dC}\sum_a p_t(a)(-\log Z_C) = \frac{1}{Z_C}\frac{dZ_C}{dC}$. The acceptance rate is $Z_C = \sum_{a \in A_C} p_p(a) + \frac{1}{C}\sum_{a \in B_C} p_t(a)$. Its derivative is:

$$\frac{dZ_C}{dC} = -\frac{1}{C^2}\sum_{a \in B_C} p_t(a).$$

Therefore, $G'(C) = \frac{1}{Z_C}\left(-\frac{1}{C^2}\sum_{a \in B_C} p_t(a)\right) \le 0$, since all terms are non-negative. The derivative is strictly negative if $\sum_{a \in B_C} p_t(a) > 0$. As $G(C)$ is continuous and piecewise differentiable with a non-positive derivative, it is non-increasing everywhere. □

**Corollary 1** (Strict Improvement over Proposal). *For any $C > 0$, OBRS produces a distribution $p_{kept,C}$ that is strictly closer to the target distribution $p_t$ than the original proposal distribution $p_p$, i.e.,*

$$KL(p_t||p_{kept,C}) < KL(p_t||p_p),$$

*unless $p_p = p_t$ or $C \le min(\frac{p_t}{p_p})$, in which case the KL divergences are both zero .*

*Proof.* From Proposition 1, we know that $\mathrm{KL}(p_t \| p_{\mathrm{kept},C})$ is non-increasing in $C$. In the limit as $C \to 0$, the acceptance probability $\alpha_C(a) \to 1$ for all $a$, meaning $p_{\mathrm{kept},C} \to p_p$. Therefore, $\lim_{C \to 0} \mathrm{KL}(p_t||p_{\mathrm{kept},C}) = \mathrm{KL}(p_t||p_p)$. For any $C > 0$, as long as $p_p \ne p_t$ and $C > min(\frac{p_t}{p_p})$, there must exist some tokens for which $p_t(a) < C \cdot p_p(a)$ or $p_t(a) > C \cdot p_p(a)$, ensuring the condition for a strictly decreasing KL divergence is met over some interval $(0,C]$. Thus, $G(C) < G(\epsilon)$ for some small $\epsilon > 0$, which implies $\mathrm{KL}(p_t||p_{\mathrm{kept},C}) < \mathrm{KL}(p_t||p_p)$. □

### A.4 PRACTICAL GUIDANCE

- **Setting $C$.** $C=1$ is a robust default: it contracts the per-prompt KL (strictly unless $p_{\mathrm{inf}} = p_{\mathrm{tr}}$), keeps acceptance high ($Z_1$), and is $O(1)$ per token. Larger $C$ pushes $q_C$ closer to $p_{\mathrm{tr}}$ but reduces throughput ($Z_C \le 1/C$); use it only if variance or bias considerations demand stronger alignment.

- **Compatibility.** The rule uses only per-token log-probabilities already computed by PPO/GRPO, so it introduces no new estimators and preserves gradient flow exactly as described in Algorithm 1.

## B DETAILS OF THE EXPERIMENTS

### B.1 TIS ADJUSTMENT EXPLANATION

However, the delay in updates results in staleness, which we magnify for evaluating our method.

In this section, we push the above scenario to its limit by asking the inference batch size to be 64x and 128x the training batch size. Concretely, we choose to use a training batch size of 32. We train models on the DeepScaleR-Preview dataset Luo et al. (2025), which contains 40k challenging competition math problems. We select Qwen3-4B-Base and Qwen3-8B-Base models Yang et al. (2025) to run RL on. An important baseline to our method is the Truncated Importance Sampling (TIS) as in Wu et al. (2025a). However, in the original technique is proposed only for the approximate models in the inference server. Thus, in the following loss form.

$$E_{a \sim \pi(\theta_{\mathrm{old}})}\left[\frac{\pi(\theta_{\mathrm{ref}})}{\pi(\theta_{\mathrm{old,inf}})}\nabla_\theta \mathrm{clip}\left(\frac{\pi(\theta_{\mathrm{new}})}{\pi(\theta_{\mathrm{ref}})}\hat{A}\right)\right]$$

The weights update frequency is close to on-policy settings, but the gap between the approximate and efficient inference model and the training weights is the primary goal to solve. However, the original formula cannot easily be adapted for our extremely large batch setting, as there is no term regularizing the difference between $\pi_{\theta_{\mathrm{new}}}$ and $\pi_{\theta_{\mathrm{old}}}$. We found that a very simple trick results in a very strong baseline on top of the TIS method, that is we write it this way.

$$E_{a \sim \pi(\theta_{\mathrm{old}})}\left[\frac{\pi(\theta_{\mathrm{new,detached}})}{\pi(\theta_{\mathrm{old,inf}})}\nabla_\theta clip\left(\frac{\pi(\theta_{\mathrm{new}})}{\pi(\theta_{\mathrm{new,detached}})}\hat{A}\right)\right]$$

Table 3: Evaluation scores across benchmarks (GSM8K, MATH-500, AMC22 & AMC23).

| Models / Methods | GSM8K | | MATH-500 | | AMC22 & AMC23 | |
|---|---|---|---|---|---|---|
| | Mean@4 | Pass@4 | Mean@4 | Pass@4 | Mean@4 | Pass@4 |
| **Qwen3-4B-Base on DeepScaler** (rollout batch size = 2048; train batch size = 32; 64×) | | | | | | |
| On Policy | 92.19 | 95.988 | 81.55 | 88.00 | 58.43 | 71.76 |
| Off Policy | 88.04 | 95.03 | 71.15 | 82.24 | 39.15 | 55.57 |
| TIS | 89.67 | 95.53 | 72.00 | 80.96 | 42.77 | 56.71 |
| TIS with Adjustment | **92.76** | **96.09** | 79.50 | 85.81 | **57.22** | **66.61** |
| Jackpot (Ours) | 92.24 | 95.891 | **80.05** | **85.89** | 53.916 | 65.034 |
| **Qwen3-4B-Base on DeepScaler** (rollout batch size = 4096; train batch size = 32; 128×) | | | | | | |
| Off Policy | 79.70 | 92.96 | 60.20 | 76.60 | 33.00 | 48.446 |
| TIS with Adjustment | 20.697 | 43.00 | 19.10 | 37.751 | 7.80 | 17.83 |
| Jackpot (Ours) | **92.00** | **95.00** | **80.00** | **85.50** | **51.20** | **60.35** |
| **Qwen3-8B-Base on DeepScaler** (rollout batch size = 2048; train batch size = 32; 64×) | | | | | | |
| On Policy | 94.238 | 96.78 | 93.99 | 96.65 | 28.95 | 37.89 |
| Off Policy | 91.05 | 95.62 | 77.15 | 84.90 | 50.60 | 65.20 |
| TS with Adjustment | 93.85 | 96.58 | 82.55 | 88.38 | 60.54 | 72.79 |
| Jackpot (Ours) | **94.01** | **96.63** | **83.05** | **88.76** | **63.55** | **74.12** |

where we use the detached most recent model output distribution as the term in the importance sampling. However, the consequence is also very clear, the internal clip around ratio is now 'short-circuited' or no longer useful. Nevertheless, the setting produces very strong convergence is correction over the off-policy baseline. We call it TIS-adjusted.

To fairly compare against the baseline, we also modify our training loss as follows, effectively also 'short-circuiting' the internal ratio clip. The only difference between our setting and theirs is that we use our proposed sampling method to regularize the $\pi_{\theta_{\text{old,inf}}}$.

$$E_{a \sim \pi(\theta_{\text{old}})} \left[ \frac{\pi(\theta_{\text{new,detached}})}{\pi(\theta_{\text{old,inf}})^{*\text{new}}} \nabla_{\theta} \text{clip}\left( \frac{\pi(\theta_{\text{new}})}{\pi(\theta_{\text{new,detached}})} \hat{A} \right) \right]$$

.

## B.2 FULL DETAILS OF EXTREME SIZE BATCH SIZE EXPERIMENTS

## C ABLATION STUDIES ON COMPONENTS OF JACKPOT, THRESHOLD, AND TOP-K

### C.1 COMPONENTS' CONTRIBUTION TO JACKPOT

Rejection bridges the gap between the rollout and the current policy (see FP8 on-policy training without explicit jackpot reweighting; this is essentially the vanilla PPO loss with the probability fixed, and the OBRS distribution in this case matches the reference policy exactly). Because the rollout–training distribution gap is now removed, stability is significantly better than the vanilla baseline: even without TIS, training does not crash.

However, without correct importance sampling (see the huge-staleness training regime, where the importance distribution no longer matches $P_{\text{ref}}$ but instead matches the current policy), training will eventually collapse.

On the other hand, jackpot reweighting does *not* solve the rollout–training gap (because under the FP8 setting it falls back to vanilla on-policy training, which again leads to a crash). But in the huge-staleness regime, where the rollout–training gap is not the main issue, jackpot reweighting combined with OBRS is effective: it performs correct importance sampling, tracks the proper target distribution, and keeps the overall training procedure stable and efficient.

### C.2 THRESHOLDS, C1, AND C2

Our method involves three hyperparameters: $C_1$, $C_2$, and the rejection threshold $\lambda$. All of them are straightforward to set, and the technique is robust across a wide range of choices.

Table 4: Evaluation scores across benchmarks (AMC12 2024, AIME24, AIME25).

| Models / Methods | AMC12 2024 | | AIME24 | | AIME25 | |
|---|---|---|---|---|---|---|
| | Mean@4 | Pass@4 | Mean@16 | Pass@16 | Mean@16 | Pass@16 |
| **Qwen3-4B-Base on DeepScaler** (rollout batch size = 2048; train batch size = 32; 64×) | | | | | | |
| On Policy | 51.11 | 65.45 | 23.12 | 33.13 | 22.91 | 30.95 |
| Off Policy | 29.44 | 41.802 | 13.958 | 23.03 | 11.042 | 18.607 |
| TIS | 31.667 | 50.844 | 11.875 | 17.561 | 11.25 | 17.278 |
| TIS with Adjustment | 43.33 | 60.67 | 18.75 | 26.03 | 17.708 | 24.607 |
| Jackpot (Ours) | **50.00** | **63.00** | **20.625** | **29.484** | **18.125** | **23.627** |
| **Qwen3-4B-Base on DeepScaler** (rollout batch size = 4096; train batch size = 32; 128×) | | | | | | |
| Off Policy | 24.44 | 38.41 | 8.00 | 15.73 | 5.00 | 11.00 |
| TIS with Adjustment | 5.00 | 11.00 | 1.00 | 4.00 | 1.00 | 2.00 |
| Jackpot (Ours) | **47.22** | **57.99** | **19.16** | **24.58** | **18.52** | **25.078** |
| **Qwen3-8B-Base on DeepScaler** (rollout batch size = 2048; train batch size = 32; 64×) | | | | | | |
| On Policy | 54.44 | 68.47 | 28.95 | 37.89 | 22.50 | 28.542 |
| Off Policy | 40.00 | 54.75 | 18.54 | 28.67 | 14.16 | 21.98 |
| TS with Adjustment | 48.33 | 59.78 | 24.58 | 35.06 | 20.00 | 22.90 |
| Jackpot (Ours) | **54.44** | **66.09** | **26.87** | **36.23** | **20.41** | **26.57** |

Table 5: FP8 on-policy training (no staleness). Best scores before crash for the vanilla baseline.

| **Experiments** | **AIME24** | **AMC** | **MATH500** | **GSM8K** |
|---|---|---|---|---|
| Vanilla (best before crash) | 23.958 | 57.530 | 80.900 | 92.665 |
| Masking-only | 25.625 | 62.048 | 83.700 | 92.835 |
| Masking & reweighting | 26.667 | 62.651 | 82.450 | 92.305 |

Table 6: BF16 training with rollout staleness (64/2048). Best scores before crash for masking-only.

| **Experiments** | **AIME24** | **AMC** | **MATH500** | **GSM8K** |
|---|---|---|---|---|
| Masking-only (best before crash) | 19.167 | 49.699 | 78.750 | 91.793 |
| Masking & reweighting | 25.625 | 63.855 | 83.800 | 92.400 |

$C_1$. We follow standard truncated importance sampling (TIS) choices. Empirically, selecting $C_1 \in [2,10]$ consistently works well, and the method is not sensitive within this interval.

$C_2$ **(upper bound for $p_{\theta_{\mathrm{ref}}}/p_{\theta_{\mathrm{new}}}$).** This parameter has no practical effect on performance. We set $C_2$ slightly larger than $1+\varepsilon_{\mathrm{high}}$ (e.g., 1.28 for DAPO), where any ratio clipped by $C_2$ would already be clipped by the PPO trust region. Thus, $C_2$ mainly serves as a conceptual safeguard for ratio stability.

**Rejection threshold $\lambda$.** Our method performs well across all experiments with a default setting of $\lambda = 1.0$, and we recommend choosing $\lambda$ close to this value. Increasing $\lambda$ makes the kept-token distribution closer to the target policy but increases the rejection rate. If $\lambda > 1$, it begins rejecting tokens even when the policy and inference distributions are perfectly aligned, causing overly conservative updates. The default value $c = 1.0$ guarantees full acceptance in the matched-distribution case and already reduces KL substantially while keeping a high acceptance rate.

**Summary.** Jackpot is stable and easy to configure: $C_1$ is robust within the typical TIS range $2-10$, $C_2$ has no practical impact once chosen above $1+\varepsilon_{\mathrm{high}}$, and $c = 1.0$ serves as a reliable default.

Table 7: Effect of $C_1$ on benchmark performance. **Experiment Setup:** Model: Qwen3-4B-Base, C2 = 3.0, threshold $c=1.0$, response limit: 8k, mini-batch/train-batch: 64/2048, PPO clip: 0.4/0.7, 100k examples. Numbers are pass@1 accuracy.

| Hyperparameters | AIME24 | AMC | MATH500 | GSM8K |
|---|---|---|---|---|
| $C_1=2$ | 26.875 | 63.855 | 82.800 | 92.267 |
| $C_1=3$ | 25.625 | 63.855 | 83.800 | 92.703 |
| $C_1=4$ | 26.042 | 65.060 | 83.500 | 92.684 |
| $C_1=8$ | 26.875 | 63.253 | 83.100 | 92.437 |

Table 8: Effect of rejection threshold $c$ on benchmark performance. **Experiment Setup:** Model: Qwen3-4B-Base (target) / Qwen3-1.7B-Base (rollout); Generation length limit: 8K; Training examples: 9K. Numbers are pass@1 accuracy.

| Threshold $c$ | AIME24 | AMC | MATH500 | GSM8K |
|---|---|---|---|---|
| 0.8 | 14.7 | 49.4 | 74.5 | 92.0 |
| 0.9 | 12.5 | 47.0 | 74.6 | 91.8 |
| 1.0 | 14.7 | 48.5 | 74.6 | 92.2 |
| 1.1 | 12.3 | 47.4 | 74.1 | 92.0 |
| 1.2 | 13.5 | 45.8 | 74.1 | 91.9 |

## C.3 CHOICE OF TOP-$K$ FOR $Z$ APPROXIMATION

In Section 4.4 we approximate the OBRS normalization constant $Z$ by summing over the union of the top-$k$ tokens under $p_{\text{inf}}$ and $p_{\theta_{\text{new}}}$, yielding $Z_{\text{approx}} \leq Z$. Increasing $k$ strictly improves this approximation but also increases the number of logits that must be materialized and stored. In the main experiments we therefore fix $k=20$, and here we justify this choice empirically.

We first study the direct effect of $k$ on the quality of the $Z$ estimator. For the extreme off-policy configuration, we log (i) the fraction of the true normalization captured by the top-$k$ estimator, $Z_{\text{approx}}/Z$, and (ii) the calibration factor $\kappa$ defined in Section 4.4.2, using $k \in \{10, 20, 40\}$. As expected, larger $k$ improves both quantities, but with rapidly diminishing returns. Even for the smallest value $k=10$ we already capture at least $87\%$ of $Z$ at the very beginning of training, and more than $99.98\%$ once the policy has warmed up within few steps; using $k=40$ increases the captured mass only slightly (from roughly $91\%$ initially to about $100.0\%$ in the steady state). Since the union of top-20 tokens is always a superset of the union of top-10 tokens, these diagnostics imply that $k=20$ already yields an almost exact estimator of $Z$ while keeping the additional overhead modest.

We next evaluate how $k$ affects downstream performance. Table 9 reports pass@1 accuracy on our four math benchmarks for Jackpot with $k \in \{10, 20, 40\}$, keeping all other hyperparameters fixed. The differences across choices of $k$ are small and non-monotonic: $k=20$ performs slightly better on AMC and MATH500, while AIME24 and GSM8K show no consistent trend. Overall, the variation is comparable to run-to-run noise, and there is no evidence that pushing $k$ beyond 20 systematically improves task performance. Taken together,

Table 9: Effect of top-$k$ on benchmark performance. Numbers are pass@1 accuracy.

| Hyperparameters | AIME24 | AMC | MATH500 | GSM8K |
|---|---|---|---|---|
| $k=10$ | 28.958 | 61.446 | 82.650 | 92.608 |
| $k=20$ | 25.625 | 63.855 | 83.800 | 92.703 |
| $k=40$ | 27.083 | 61.446 | 83.200 | 92.418 |

these results show that larger $k$ does improve the $Z$ estimator, but the gains become marginal once $k$ reaches 20. Since the computational overhead of our method scales roughly linearly with $k$, we adopt $k=20$ as a practical default: it provides an accurate, well-calibrated estimate of $Z$ with negligible additional cost (less than $3\%$ overhead in our setup), and larger values of $k$ do not yield measurable benefits in our experiments.

