# OpenReview forum: "Jackpot: Align Actor-Policy Distribution for scalable and stable RL for LLM"
_ICLR.cc/2026/Conference — ICLR 2026 Poster_

### Official Review · Reviewer_k9JV · 2025-10-29

**Soundness:** 3
**Presentation:** 3
**Contribution:** 2
**Rating:** 6
**Confidence:** 3

**Summary:**

JACKPOT targets, actor–policy mismatch, the PPO pain point in LLM RL. It fixes the problem at sampling time with Optimal-Budget Rejection Sampling. The naïve OBRS recipe is unusable at scale because it needs full-vocab normalizers, yields unbounded correction weights, and can kill throughput when alignment is too strong. The paper’s value is a practical pipeline: estimate OBRS on a top-k union of actor/policy tokens and calibrate the missing mass with the simple identity that the normalizer equals the expected acceptance rate; then train with a stabilized PPO objective that factorizes the correction and clips each ratio so gradients stay bounded and trust-region friendly. Empirically it demonstrates it shines at where vanilla PPO struggles, large rollout batches at fixed minibatch, and extreme off-policy collection, delivering steadier learning and better scores without changing the base model or reward loop.

**Strengths:**

1. Targets PPO’s real pain points. It directly attacks actor–policy mismatch in the regimes that break PPO such as huge rollout batches with the same small minibatch and extreme off-policy actors, and shows steadier learning and better scores there.
2 Not a naïve OBRS port. The paper surfaces why a straight plug-in fails (full-vocab normalizers, unbounded weights, acceptance-throughput collapse) and then fixes each with a practical trio: top-k union for feasibility, acceptance-rate calibration for unbiased scale, and a factorized, clipped correction that keeps updates stable.
3. Clean narrative. The writing crisply motivates the problem, diagnoses naïve failures, and walks the reader through the recipe and its effects; the algorithm is easy to implement from the description.

**Weaknesses:**

1. Loss clipping bias. The stabilized objective clips two likelihood ratios to keep gradients tame. That’s an intentional bias; it can underweight rare-but-informative tokens and shrink effective step size. The paper argues stability, but it does not quantify when the bias alters policy improvement or exploration.
2. Actor–policy visitation skew. OBRS keeps tokens the current policy already likes, which shifts the behavior distribution. Alignment lowers variance, but it can also prune rare, hard states that matter for learning. Without coverage/diversity diagnostics, it is unclear whether OBRS quietly narrows what the policy ever sees.
3. Sequence or block-level OBRS. Tokenwise rejection is simple, but it fragments credit on long sequences where decisions cohere over spans. Span- or block-level acceptance could improve temporal coherence at the same keep-rate (see questions below).

**Questions:**

1. Clipping bias: compare unclipped vs clipped vs two-sided clipped weights on identical runs; report policy-improvement proxies, gradient norms, collapse rate.
2. Visitation skew: show state-coverage heatmaps before/after OBRS, effective sample size per update, entropy of the kept policy, and advantage-weighted coverage of rare regions.
3. Top-k curve: plot performance, gradient norms, and normalizer error versus k; provide a simple k rule or a dynamic k policy tied to a target acceptance band.
4. Potential KL "double counting"?: With OBRS plus the clipped reference ratio acting like a trust region, an explicit KL(pθ‖pref) may over-constrain updates. add curves with and without the explicit KL term to the reference under OBRS; clarify whether the second clipped ratio already suffices or if a reduced KL is still helpful.
5. Curious to see block-level OBRS.

---

> ### Author Response · Authors · 2025-11-21
> **Response to Reviewer k9JV (Part 1)**
>
> We thank the reviewer for the thoughtful review and constructive suggestions. We are glad that the reviewer found our work **targeting the pain points of PPO** and that it has a **clean narrative**. We have carefully addressed your questions. We hope the reviewer will consider raising your score in light of our response.
>
> ## Q1 Visitation skew
>
> We are aware that high-entropy tokens are essential for RL training [1].
>
> Our observation shows the entropy of the kept policy and rollout is almost identical (**difference < 3e-3**). These observations suggest that OBRS does not introduce harmful visitation skew.
>
> **Experiment Setups**
>
> *Llama-3.2-3B-Instruct/Qwen3-4B-Base with a large batch size (2048 examples per batch).*
>
> | Model                | Rollout entropy | Kept entropy |
> |----------------------|-----------------|--------------|
> | Llama3.2-3B-Instruct | 1.521           | 1.520        |
> | Qwen3-4B-Base        | 0.85            | 0.84         |
>
> *Entropy in training step 10*
>
> [1] https://arxiv.org/pdf/2506.01939
>
> ## Q2 Sequence or Block OBRS
>
> Our current implementation does not show a significant difference between sequence-level and token-level rejections.  We leave this as an interesting future work.
>
> **Experiment Setup**
>
> *Model: Qwen3-4B-Base, C1=C2=3.0, threshold c=1.0, response limit: 8k, mini batch/train batch: 64/2048, ppo clip: 0.4/0.7, 100k examples*
>
> | Hyper-parameters | AIME24 | AMC    | MATH500 | GSM8K  |
> |------------------|--------|--------|---------|--------|
> | Seq (geo mean)   | **26.25**  | 62.952 | 83.05   | 92.286 |
> | token            | 25.625 | **63.855** | **83.8**    | **92.703** |
>
> ## Q3 Clipping Bias
>
> We implement two clipping in Jackpot, namely $C_1$ and $C_2$. The two clipping terms only introduce negligible bias under our setting.  We report the clipping rate of the typical training settings:
>
> **Experiment Setups**
>
> *Model: Qwen3-4B-Base, C1=C2=3.0, threshold c=1.0, response limit: 8k, mini batch/train batch: 64/2048, ppo clip: 0.4/0.7*
>
> |      | First clip | Second clip |
> |------|------------|-------------|
> | Bf16 | 0.0        | 0.0         |
> | Fp8  | 5.05e-4    | 0.0         |
>
> - $C_1$ **(seldom triggered)** regulates the upper bound on the ratio $\frac{p_{\theta_{\mathrm{new}}}}{p_{\theta_{\mathrm{inf}}}}$ preventing situations in which $p_{\theta_{\mathrm{inf}}}$ is extremely small while $p_{\theta_{\mathrm{new}}}$ is not, which would otherwise cause this ratio to become unexpectedly large. In practice, however, such cases are very rare because rollouts are sampled directly from $p_{\theta_{\mathrm{inf}}}$, making tokens with extremely small $p_{\theta_{\mathrm{inf}}}$ unlikely to appear. Our empirical results indicate that, almost no tokens (5e-4) are clipped by $C_1$. We enable $C_1$ clip as an implementation only to avoid the extreme case.
>
>
> - $C_2$ **(never triggered)** is the upper bound for $p_{\theta_{\mathrm{new}}} / p_{\theta_{\mathrm{ref}}}$. This parameter has no practical effect on performance. We set $C_2$ slightly larger than $1 + \varepsilon_{\mathrm{high}}$ (e.g., $1.28$ for DAPO), where any ratio clipped by $C_2$ would already be clipped by the trust region used in PPO. Thus, $C_2$ mainly serves as a conceptual safeguard for ratio stability.
>
>
> ## Q4 Top-k curve
>
> Existing results show that larger k does improve the Z estimator, but the gains become marginal once k reaches 20.
>
> **Experiment Setups**
>
> *Model: Qwen3-4B-Base, C1=C2=3.0, threshold c=1.0, response limit: 8k, mini batch/train batch: 64/2048, ppo clip: 0.4/0.7, 100k examples*
>
> | Hyper-parameters | AIME24 | AMC    | MATH500 | GSM8K  |
> |------------------|--------|--------|---------|--------|
> | k=10             | 28.958 | 61.446 | 82.650  | 92.608 |
> | k=20             | 25.625 | 63.855 | 83.800  | 92.703 |
> | k=40             | 27.803 | 61.446 | 83.200  | 92.418 |
>
>
> Even for the smallest value $k=10$ we already capture at least $87$% of $Z$ at the very beginning of training, and more than $99.98$% once the policy has warmed up within few steps; using $k=40$ increases the captured mass only slightly (from roughly $91$% initially to about $100.0$% in the steady state). Since the union of top-$20$ tokens is always a superset of the union of top-$10$ tokens, these diagnostics imply that $k=20$ already yields an almost exact estimator of $Z$ while keeping the additional overhead modest (less than $3$% overhead in our setup).
>
> We adopt k = 20 as a practical default: it provides an accurate, well-calibrated estimate of Z with negligible additional cost, and larger values of k do not yield measurable benefits in our experiments.
>
> We included the discussion of top-k in our modified paper $\\color{blue}{\\text{(Sections C.3, p.g. 18-19)}}$.

---

> ### Author Response · Authors · 2025-11-21
> **Response to Reviewer k9JV (Part 2)**
>
> ## Q5 Potential KL "double counting"
>
> We do not have the KL double-counting problem. Our method **does not introduce any explicit KL loss** between
> $p_{\theta_{\mathrm{new}}}$ and $p_{\theta_{\mathrm{ref}}}$, **nor do we apply additional
> regularization of this ratio**.
>
> This design follows directly from our implementation of the $C_2$ parameter. In practice, we set $C_2$ to be slightly larger than $1 + \varepsilon_{\mathrm{high}}$ (e.g., $1.28$ for DAPO). Under this choice, any ratio $p_{\theta_{\mathrm{ref}}} / p_{\theta_{\mathrm{new}}}$ that $ C_2$ would clip is already clipped by the underlying PPO  trust region. As a result, the upper bound $C_2$ **never becomes active** during training; it is included only to conceptually indicate that the ratio is bounded, while the trust-region mechanism provides the actual stabilization. The reference ratio is a component of the Importance Sampling ratio, which is required in PPO-style losses.
>
> **Our effective restriction on $p_{\theta_{\mathrm{ref}}} / p_{\theta_{\mathrm{new}}}$ is therefore always aligned with the original trust-region limitation used in PPO, without redundant penalties.**. Hence, we do not have the KL double-counting problem.

---

### Official Review · Reviewer_43dn · 2025-10-31

**Soundness:** 2
**Presentation:** 3
**Contribution:** 2
**Rating:** 4
**Confidence:** 3

**Summary:**

The paper introduces a new reinforcement learning framework aimed at improving the stability and efficiency of RL-based fine-tuning for large language models. The central problem addressed is the distribution mismatch between the actor (rollout) model and the policy being optimized, which often arises in large-batch, asynchronous, or off-policy training scenarios. Existing importance sampling corrections for this mismatch tend to suffer from instability or poor performance. To tackle this, the authors propose JACKPOT, a method that leverages Optimal Budget Rejection Sampling (OBRS) to directly reduce the KL divergence between the actor and policy distributions.

Empirical results on large-scale language models (Qwen3-4B and Qwen3-8B) demonstrate 20% improvement on AMC and 8% improvement on AIME benchmarks under extreme off-policy settings, as well as significant gains in large-batch asynchronous RL training. The method substantially delays training collapse and outperforms prior off-policy RL methods such as Truncated Importance Sampling.

**Strengths:**

1. The paper targets a concrete and important practical bottleneck: distribution mismatch between rollout models and the policy used for updates in RL for LLMs (large-batch, asynchronous, quantized/distilled rollouts, etc.). The motivation is clearly stated and practically relevant.
2. The paper proposes using Optimal Budgeted Rejection Sampling to directly reduce actor–policy KL, which is a principled move. The paper provides theoretical guarantees: OBRS is the unique budget-optimal token-wise accept/reject rule minimizing KL for a given acceptance budget, and the post-rejection distribution monotonically approaches the target as the scale parameter varies. These formal results strengthen the core idea.

**Weaknesses:**

1. The OBRS acceptance probability for a sampled token requires evaluating the ratio $p_{\theta_{new}} (a) / p_{inf} (a)$. In practice, this implies scoring candidate tokens under the *current policy* during data collection or otherwise obtaining these probabilities. However, the paper does not fully quantify the extra compute this incurs in real serving setups. This is important because the whole motivation is improving throughput; any extra compute could reduce or eliminate the gains.
2. The stabilized weight uses two clipped ratios and a stop-gradient on the alignment factor, trading bias for variance. The paper argues this is necessary for stability, but there is limited theoretical quantification of the bias introduced by clipping and the stop-gradient, nor diagnostics showing how much bias affects final performance in less extreme settings. More ablation (e.g., varying c1,c2, removing stop-grad) would help understand the tradeoffs.
3. The final stabilized Jackpot objective (Section 4.4) is a pragmatic heuristic. The paper admits that it "trades a small amount of bias (from clipping and approximation) for a massive reduction in variance". This factorization and clipping move the method away from the clean theoretical guarantees of OBRS. The paper does not provide any theoretical analysis of how this specific, biased re-weighting scheme affects the optimization landscape, convergence, or the final policy.

**Questions:**

1. In Algorithm 1, the acceptance probability $\alpha$ requires access to $p_{\theta_{new}}$ at sampling time. How is this computed in practice when rollouts are generated by a separate inference actor? Are logits from $p_{\theta_{new}}$ queried synchronously (which could double inference cost), or are they approximated, cached, or delayed? Please clarify how this fits into a large distributed RLHF or asynchronous setup, and quantify the extra compute or communication cost.
2. The stabilized Jackpot loss uses two clipped ratios and a stop-gradient. Could you formalize or empirically show how these choices affect gradient bias and variance? Specifically, how do c1,c2 and the stop-gradient operation trade off between bias and stability? It would help to include plots of gradient variance before and after stabilization.
3. The method is validated primarily on mathematical benchmarks (AMC, AIME, GSM8K, MATH-500). Evaluating on alignment or preference-based tasks (e.g., Anthropic-HH) would help test generality and robustness beyond numeric reasoning.

---

> ### Author Response · Authors · 2025-11-21
> **Official Comment to 43dn(part 1)**
>
> We thank the reviewer for the thoughtful review and constructive suggestions. We are glad that the reviewer found our work targeting concrete and important bottleneck, principled, with theoretical guarantee. We have carefully addressed your questions. We hope the reviewer will consider raising your score in light of our response.
>
>
> ## Q1 Clipping Affection
>
> We implement two clipping in Jackpot, namely $C_1$ and $C_2$.
>
> - $C_1$ (seldom triggered) regulates the upper bound on the ratio $\frac{p_{\theta_{\mathrm{new}}}}{p_{\theta_{\mathrm{inf}}}}$ preventing situations in which $p_{\theta_{\mathrm{inf}}}$ is extremely small while $p_{\theta_{\mathrm{new}}}$ is not, which would otherwise cause this ratio to become unexpectedly large. In practice, however, such cases are very rare because rollouts are sampled directly from $p_{\theta_{\mathrm{inf}}}$, making tokens with extremely small $p_{\theta_{\mathrm{inf}}}$ unlikely to appear. Our empirical results indicate that, almost none tokens (<5e-4) are clipped by $C_1$. We enable $C_1$ clip as a implementation only to avoid extreme case.
> - $C_2$ (never triggered) is the upper bound for $p_{\theta_{\mathrm{new}}} / p_{\theta_{\mathrm{ref}}}$ This parameter has no practical effect on performance. We set $C_2$ slightly larger than $1 + \varepsilon_{\mathrm{high}}$ (e.g., $1.28$ for DAPO), where any ratio clipped by $C_2$ would already be clipped by the trust region used in PPO. Thus, $C_2$ mainly serves as a conceptual safeguard for ratio stability.
>
>
> We report the clipping rate of the typical training settings:
>
> **(Model: Qwen3-4B-Base, C1=C2=3.0, threshold c=1.0, response limit: 8k, mini batch/train batch: 64/2048, ppo clip: 0.4/0.7)**
> | |First clip|Second clip
> |---|---|---|
> |Bf16|0.0|0.0|
> |Fp8|5.05e-4|0.0|
>
>
> ## Q2 Clipping Ratios and Stop Gradient
>
> - **$C_1$**: We follow the standard truncated importance sampling choices. Empirically, we find that selecting $C_1$ in the range $[2, 10]$ consistently works well, and the method is not sensitive within this interval.
>
> - **$C_2$ (upper bound for $p_{\theta_{\mathrm{ref}}} / p_{\theta_{\mathrm{new}}}$)**: This parameter has no practical effect on performance. We set $C_2$ slightly larger than $1 + \varepsilon_{\mathrm{high}}$ (e.g., $1.28$ for DAPO), where any ratio clipped by $C_2$ would already be clipped by the trust region used in PPO. Thus, $C_2$ mainly serves as a conceptual safeguard for ratio stability.
> - **Stop Gradient**:  We use the current policy $p_{\mathrm{new}}(\theta)$ to perform the OBRS. Without a stop gradient,  we will allow backpropagation through the term $p_{\mathrm{new}}(\theta)$ in the OBRS. However, since there already exists one term $p_{\mathrm{new}}(\theta)$ in the original PPO objective,  this would introduce multiple occurrences of $p_{\mathrm{new}}(\theta)$ in the backward pass, which can produce incorrect gradient signals.
>
>
> We included the discussion of clipping ratios in our modified paper $\\color{blue}{\\text{(Sections C.2, p.g. 17-18)}}$.
>
> The following table shows the final training results for different $C_1$. Our method is not sensitive to the selection of hyper-parameters.
>
> **(Model: Qwen3-4B-Base, C2=3.0, threshold c=1.0, response limit: 8k, mini batch/train batch: 64/2048, ppo clip: 0.4/0.7, 100k examples)**
> Hyper-parameters|AIME24|AMC|MATH500|GSM8K
> ---|---|---|---|---
> C1=2|26.875|63.855|82.8|92.267
> C1=3|25.625|63.855|83.8|92.703
> C1=4|26.042|65.06|83.5|92.684
> C1=8|26.875|63.253|83.1|92.437

---

> ### Author Response · Authors · 2025-11-21
> **Official Comment to 43dn(part 2)**
>
> ## Q3 Additional compute or communication costs
>
> Jackpot integrates naturally into large-scale RL systems and introduces only minimal
> computational overhead. We include a discussion of implementation and overhead in our modified paper $\\color{blue}{\\text{(Section 4.5, p.g. 7)}}$.
>
>
> To summarize,  our method do not need to calculate the token probability under current policy during generation/rollout. Instead, after rollouts, the current-policy probabilities $p$ are obtained via a single prefill pass of the target model. This prefill is already present in Verl and similar RL pipelines, where the policy model recomputes the log-probability for each trajectory. (https://github.com/volcengine/verl/blob/1b8645db79c7512804ae46080f44c5607e6bd92a/docs/workers/fsdp_workers.rst#L66) Thus, no extra trajectories are needed and no synchronous queries to the target model are introduced.
>
>
> We present a brief analysis of the overhead of Jackpot.
> ### Overhead Analysis
> Jackpot process relies on two components, both of which are inherent to the training pipeline or negligible in cost:
> 1.  Actor  Probabilities ($p_{\text{inf}}$): The rollout engine (e.g., vLLM) computes logits to generate tokens. We simply request the top-$K$ log-probabilities (typically $K=20$) for our normalization estimator during this step. Extracting top-$K$ values is a standard, low-cost operation in modern inference engines and adds no significant overhead to the autoregressive generation. (https://docs.vllm.ai/en/v0.8.2/api/inference_params.html)
>
> 2.  Policy Probabilities ($p_{\theta_{\text{new}}}$): In standard PPO/GRPO, the training engine already performs a forward pass on the collected trajectories to compute the "new" log-probabilities for the policy ratio $r_t(\theta) = \pi_{\text{new}} / \pi_{\text{old}}$. Jackpot reuses these exact logits to compute the importance weights. (https://github.com/volcengine/verl/blob/1b8645db79c7512804ae46080f44c5607e6bd92a/docs/workers/fsdp_workers.rst#L66)
>
>
> Below, we use an experiment to demonstrate the practical efficiency of Jackpot, listing the most significant time costs per RL step.
>
> Experiment Setup
>
> **(Model: Qwen3-4B-Base, C1=C2=3.0, threshold c=1.0, response limit: 8k, mini batch/train batch: 64/2048, ppo clip: 0.4/0.7, 16 * H100)**
>
> Metric|Timing (s)
> ---|---:
> Time per step|789.5
> Additional compute per step|19.6

---

### Official Review · Reviewer_BCWN · 2025-11-01

**Soundness:** 2
**Presentation:** 2
**Contribution:** 2
**Rating:** 4
**Confidence:** 4

**Summary:**

The paper aims to reduce the high rollout cost from large trajectories in reinforcement learning based post-training of LLMs while improving stability and scalability. The key idea is to sample trajectories from a lightweight actor policy different from the main training policy to make RL more efficient. To handle the resulting mismatch, the paper introduces a rejection-sampling–based approach that directly reduces the distribution gap between actor and policy. It further proposes a Top-K approximation with bias correction and a stabilized PPO loss to maintain trust-region stability. Experimental ablations shows promising gain over baselines.

**Strengths:**

- The idea of using simple light-weight policy for sampling trajectories is interesting and effective especially for larger policies
- bounding the KL and using rejection sampling helps in mitigating the distribution shift
- efficient Top-K probability estimation keeps it usable for large vocabularies.
- empirical performance are promising especially on AMC and AIME benchmarks compared to off-policy baselines.

**Weaknesses:**

- Although interesting, the idea is extremely similar to several parallel streams including speculative decoding, on policy distillation, weak-strong etc and the key novelty of the approach is not clear
- The paper ensures the KL divergence between the actor and policy distribution, however how it ensures closeness to the reference policy based on which standard RLHF policies are trained? How do you ensure closeness to that? Can you show a KL plot with the reference?
- It requires sampling multiple trajectories and computing the ratio of the probabilities, how much time it incurs additionally during training?
- Also, what kind of light-weight models can help to mitigate this shift in an efficient way - ie does compression or smaller finetuned models or distilled models, how are different models behaviour
- How sensitive is the approach with the threshold and how is the threshold determined? That will also affect the number of trajectories to generate? It will be helpful to provide the details.

**Questions:**

See Weakness.

---

> ### Author Response · Authors · 2025-11-21
> **Official comment to BCWN (Part 1)**
>
> We thank the reviewer for the thoughtful review and constructive suggestions. We are glad that the reviewer found our idea is interesting, useful with promising evaluation results. We have carefully addressed your questions. We hope the reviewer will consider raising your score in light of our response.
>
> ## Q1: Difference from parallel streams
>
> **Speculative Decoding.**
>
> *Differences*:
> a. Speculative Decoding operates at inference time and aims for an exact distribution match. It requires the target policy to run alongside the draft model in real-time to verify tokens. Crucially, when a draft token is rejected, the system must halt and trigger the target model to regenerate a new token. $\\color{blue}{\\text{(Section 4.5, p.g. 7)}}$
>
> b. Jackpot, in contrast, operates as a post-hoc correction applied during the training step. We do not interrupt the rollout or regenerate tokens; the inference engine generates the full sequence without overhead. The rejection sampling occurs later, during the standard training "forward" pass, where we simply mask rejected tokens in the gradient computation.
>
> *Summary of Advantages*:
> a. **Negligible Compute Overhead.** Unlike speculative decoding, which requires expensive extra compute for verification and regeneration, Jackpot incurs virtually zero additional cost because it leverages computations already performed for the RL loss. Since RL algorithms (like PPO/GRPO) already require computing the target policy's log-probabilities to calculate the loss, Jackpot simply reuses these values—alongside rollout log-probabilities efficiently retrieved via top-k to compute the rejection mask.
>
> b. **Strategic Alignment.** While we do not enforce an identical distribution match, Jackpot provably and significantly reduces the KL divergence between the rollout and target distributions. It offers a highly efficient trade-off: achieving the substantial distribution alignment necessary for stability without the prohibitive latency and compute costs of running the target model during rollout.
>
> **On-Policy Distillation.**
>
> 1. On-policy distillation typically optimizes a reverse-KL objective and is primarily designed for training **a smaller "student" model to mimic a larger "teacher" model.**
>
> 2. Jackpot. In contrast, our method uses reinforcement learning (based on GRPO) and targets the efficient **training of large models themselves**, rather than compressing/distilling them into a smaller model. Our focus is on improving the efficiency and stability of optimization for large policy models using extreme off-policy data.
>
> ## Q2 KL plot with the reference
>
> Our method is compatible with the standard KL-regularized RLHF setup.
>
> **Jackpot** can ensure closeness to either the **reference policy** or the **latest policy**, depending on which distribution is chosen as the target $p$ in the OBRS procedure.
> If $p$ is set to the reference policy, OBRS aligns the actor directly toward the **reference** model. In practice, the reference in RLHF training is typically the model from the previous training batch, whose logits are included in the current batch.
> If $p$ is set to the **newest** policy, OBRS aligns the actor toward the most recent policy.
>
> Empirically, we found that aligning with the newest policy provides better end-to-end training results. We clarify this in our modified paper $\\color{blue}{\\text{(Section 4.3, p.g. 6)}}$.
>
> ## Q3 Threshold
>
> 1. **Trajectory**: The rejection threshold does **not** affect the number of trajectories that will be generated. Our method never resamples rejected tokens; instead, rejected tokens are simply masked during gradient computation. Therefore, the rollout length and trajectory count remain the same as in standard RL training.
>
> 2. Threshold selection. We recommend choosing a value **close to**  $1.0$. We recommend choosing $c$ close to this value.
> Increasing c makes the kept-token distribution closer to the target policy, but also increases the rejection rate. If the threshold is set above $1$, it will start rejecting tokens even when the policy and inference distributions are perfectly aligned, leading to overly conservative updates.
>
> 3. The default setting ($c = 1.0$) guarantees full acceptance in the matched-distribution case and, in our experiments, already reduces the KL substantially while maintaining a high acceptance rate.
> In the table below, we report the effect of different thresholds to illustrate the method’s **insensitivity** to this choice (when we choose $c$ close to 1). We clarify this in our modified paper $\\color{blue}{\\text{(Section C.2, p.g. 17-18)}}$.
>
> **Experiment Setup**
>
> Model: Qwen3-4B-Base (target)/Qwen3-1.7B-Base (rollout)
> Generation length limits: 8K
> Total Training Examples: 9K
> | Threshold | AIME24 |  AMC | MATH500 | GSM8K |
> |-:|-:|-:|-:|-:|
> |0.8|14.7|49.4|74.5|92.0|
> |0.9|12.5|47.0|74.6|91.8|
> |1.0|14.7|48.5|74.6|92.2|
> |1.1|12.3|47.4|74.1|92.0|
> |1.2|13.5|45.8|74.1|91.9|

---

> ### Author Response · Authors · 2025-11-21
> **Official comment to BCWN (Part 2)**
>
> ## Q4: additional time during training
> Our method **does not** require sampling more trajectories than standard RL training. The importance ratios are obtained directly during rollout: the behavior-policy probabilities $q$ are computed by the rollout engine (e.g., vLLM) and returned together with the sampled tokens, and the target-policy probabilities $p$ are obtained via a single prefill pass of the target model. This prefill is already present in Verl and similar RLHF pipelines, where the policy model recomputes the log-probability for each trajectory.
>
> Therefore, our approach introduces almost **no extra sampling cost** and essentially **no additional inference overhead** relative to standard RL setups. We include a discussion of implementation and overhead in our modified paper $\\color{blue}{\\text{(Section 4.5, p.g. 7)}}$. To summarize,
>
> **Overhead Analysis**
> Jackpot process relies on two components, both of which are inherent to the training pipeline or negligible in cost:
> 1.  **Actor  Probabilities ($p_{\text{inf}}$)**: The rollout engine (e.g., vLLM) computes logits to generate tokens. We simply request the top-$K$ log-probabilities (typically $K=20$) for our normalization estimator during this step. Extracting top-$K$ values is a standard, low-cost operation in modern inference engines and adds no significant overhead to the autoregressive generation.
> 2. **Policy Probabilities ($p_{\theta_{\text{new}}}$)**: In standard PPO/GRPO, the training loop already performs a forward pass on the collected trajectories to compute the "new" log-probabilities for the policy ratio $r_t(\theta) = \pi_{\text{new}} / \pi_{\text{old}}$. Jackpot reuses these exact logits to compute the importance weights.
>
> Therefore, our method requires no extra inference passes and almost zero additional training time relative to standard RL setups.
>
> Here is an example:
>
> **Experiment Setup**
>
> **(Model: Qwen3-4B-Base, C1=C2=3.0, threshold c=1.0, response limit: 8k, mini batch/train batch: 64/2048, ppo clip: 0.4/0.7, 16 * H100)**
> | Metric                      | Timing (s) |
> |-----------------------------|-----------:|
> | Time per step               | 789.5      |
> | Additional compute per step | 19.6       |
>
> ## Q5 Lightweight Models
>
> We explored **three types** of models to assess how effectively they can mitigate distribution shift while keeping rollout generation efficient:
>
> 1. **Staled models.**  These models lag behind the current policy by $2$--$4$k training examples and tens of updates. They maintain high acceptance rates and result in stable training. In our experiments, their final performance nearly matches that obtained with the original policy model during rollouts.
>
> 2. **Compressed models (e.g., FP8 KV).**  These models share the same architecture series as the policy model but use lighter-weight quantized KV. They also achieve high acceptance rates and enable stable RL training, with performance close to that of rollouts generated by the original model.
>
> 3. **Smaller models from the same model family.**  The acceptance rate varies by model pair.  \textsc{Jackpot} significantly improves training efficiency and stability. However, the final performance still depends on the compatibility between the rollout model and the target policy.  Overall, several lightweight rollout models can be effective, provided their acceptance rate relative to the target policy is reasonably high. Determining how to construct or select an optimal lightweight rollout model reliably remains an open question, and we regard this as promising future work.

---

> > ### Comment · Reviewer_BCWN · 2025-11-28
> > **Response to Rebuttal by Authors**
> >
> > I thank the authors for the detailed rebuttal and appreciate the responses and ablations which clears most of my concerns.
> > However, i am slightly confused as the authors mention no additional cost and no additional samples during training.
> > So to clear my doubt, method generates multiple trajectories from simple light-weight policy and select the ones which are close to the on-policy right? So one needs to generate much larger trajectories during training from the light-weight policy?

---

> > > ### Author Response · Authors · 2025-11-28
> > >
> > > Thank you for the question.
> > >
> > > To clarify: our method does **not** require generating more trajectories than standard RL training.
> > >
> > > Our key point is that Optimal Budget Rejection Sampling (OBRS) operates at the **token level**, not the trajectory level. During training, we generate **the same number** of trajectories using the lightweight policy (via vLLM), precisely as in standard RL training. *We do not generate a larger pool of trajectories to pick from.*
> > >
> > > After generation, for each produced trajectory we apply the masking strategy described in $\\color{blue}{\\text{(Section 4.1, p.g. 5)}}$ and $\\color{blue}{\\text{(Figure 3, p.g. 6)}}$: we selectively mask tokens rejected by our trained policy model (the model we actually want to train). These masked tokens do not participate in gradient computation or backpropagation, which means the FSDP engine does not process them.
> > >
> > > In other words， the trajectory generation cost is **unchanged (same as standard training)**. No enlarged candidate set is produced. The only difference is that some tokens inside each trajectory are dropped from optimization (and the kept tokens are reweighted, in $\\color{blue}{\\text{(Section 4.1, p.g. 5)}}$).
> > >
> > > Thus, our method does not increase trajectory generation, and it introduces no additional sampling overhead.

---

### Official Review · Reviewer_ZoQc · 2025-11-01

**Soundness:** 3
**Presentation:** 2
**Contribution:** 2
**Rating:** 4
**Confidence:** 3

**Summary:**

This paper addresses the computational cost and instability arising from actor-policy distribution mismatch in reinforcement learning for large language models. While allowing these distributions to differ could enable significant efficiency gains—such as large-batch training, asynchronous updates, or using smaller rollout models—existing importance sampling-based corrections face a fundamental trade-off between stability and performance. The authors propose Jackpot, which leverages Optimal Budget Rejection Sampling (OBRS) to directly reduce the distributional gap between actor and policy networks. The method introduces three key contributions: an OBRS-based masking mechanism that maintains closer alignment between distributions, an efficient probability estimation strategy using Top-K logits with batch-wise bias correction to handle memory constraints, and a stabilized PPO loss jointly accounting for importance sampling ratios and trust-region constraints. Evaluated on large-batch training (128 mini-batches per rollout) and extreme off-policy scenarios, Jackpot demonstrates 20% improvement on AMC benchmarks and 8% on AIME benchmarks over baseline methods while maintaining stable training dynamics under severe distributional mismatch conditions.

**Strengths:**

- This paper tackles an important and practically relevant problem in reinforcement learning for LLMs: the gradient estimation bias and training instability caused by distribution mismatch between the actor and policy networks.

- The proposed OBRS-based approach seems to be a principled solution to directly reduce this distributional gap.

-  The empirical evaluation demonstrates promising results, particularly in maintaining training stability.

**Weaknesses:**

- The paper lacks a detailed discussion of the differences between importance sampling ratio-based corrections and the rejection sampling approach, despite both relying on importance sampling ratio calculations. This makes it unclear why the optimal rejection sampling method provides advantages over existing correction techniques like TIS.

- The paper builds upon Optimal Budgeted Rejection Sampling (OBRS) but does not provide sufficient introduction or justification for this choice. Without adequate background and motivation, it is difficult for readers to accept that OBRS is the appropriate approach for addressing the distribution mismatch problem.

- The empirical evaluation lacks comprehensive ablation studies to disentangle the contributions of individual components. It remains unclear whether the OBRS-based masking mechanism, the Jackpot re-weighting strategy, or both components are essential for achieving the reported improvements in stability and performance.

**Questions:**

- The LaTeX format of this paper does not appear to follow the standard ICLR submission template.

- Does the Phase 1 Data Collection with OBRS require modifications to the sampling code in vLLM or other inference frameworks?

- How should the hyperparameters $c_1$ and $c_2$ be chosen? Is there any guidance provided for their selection. Furthermore, is the method robust to different choices of these hyperparameters?

---

> ### Author Response · Authors · 2025-11-21
> **Official Comment to ZoQc**
>
> We thank the reviewer for the thoughtful review and constructive suggestions. We are glad that the reviewer found our work **principled** and solved **important, practical** problems, with promising evaluation results. We have carefully addressed your questions. We hope the reviewer will consider raising your score in light of our response.
>
> ## Q1 Discussion of the difference between IS ratio corrections
>
>
> The mechanisms and objectives of OBRS and importance ratio corrections (e.g., TIS) are fundamentally different and complementary rather than competing.
>
> Our approach transforms the actor distribution $q$ into a new distribution $q^*$ that is intrinsically closer to the target policy $p$. This transformation adjusts the sampling process via rejection sampling, producing data that more closely resemble samples from $p$.
>
> A key point is that this distribution-alignment step is independent of importance sampling:
> Jackpot modifies how responses are sampled, reducing the mismatch between $q$ and $p$ at the data level. In our proposed Jackpot, we first use vLLM to generate responses, then mask unaccepted tokens, eliminating them from the gradient calculation and backward pass, effectively changing the sampled token’s probability of appearance. We illustrate this process in the modified paper $\\color{blue}{\\text{(Figure 3, p.g. 6)}}$.
> IS-based correction methods operate after sampling, adjusting the importance ratio $p/q$ to reduce bias to perform estimation.
>
> Therefore, OBRS naturally complements correction techniques such as TIS. OBRS reduces the distribution mismatch by bringing $q$ closer to $p$, while TIS can be applied on top to further correct bias. With more closely matched distributions $q$ and $p$, the accuracy of importance sampling is known to be improved.
>
> Moreover, previous TIS-based methods suffer when there is a large distribution gap between actor and policy mismatch, as the importance ratio forces the trajectories sampled by the actor of low likelihood (q large, p small), causing a training-inference mismatch, as these tokens from q will very unlikely to be sampled by p alone. Thus, the issue motivates our method further, which directly adjusts and modifies the actor’s sampling tokens.
>
> We clarify this in the related work of the modified paper $\\color{blue}{\\text{(Section 2.2, p.g. 3-4)}}$.
>
> ## Q2 Background and Motivation of OBRS
>
> Thanks for the insightful question. We modified our paper to discuss the background and motivation of using OBRS $\\color{blue}{\\text{(Section 3, p.g. 4-5)}}$.
>
> **Why is rejection sampling a natural tool for distribution mismatch?**
> Rejection sampling provides a classical, direct mechanism for reshaping the sampling distribution itself, rather than relying solely on post-hoc importance ratio corrections (which TIS performs). By selectively accepting samples according to a function of $p/q$, rejection sampling naturally shifts the effective sampling distribution toward regions preferred by $p$, making it well-suited for mitigating distribution mismatch and stabilizing importance ratios when used with IS-based methods.
>
>
> **Why OBRS instead of naive rejection sampling?**
> OBRS is the optimal rejection sampling strategy under an acceptance-rate constraint. OBRS enjoys several properties that directly address the challenges posed by off-policy distribution mismatch:
>
> - **(Rejection Ratio)** The LLM's sampling distribution is extremely high-dimensional, with hundreds of thousands of vocabulary items, resulting in the constant C used in rejection sampling being very large; hence, naive rejection sampling will reject almost all tokens generated by the actor, since most token probabilities are exceedingly close to zero.
>
> - **(Theoretical Guarantee)** Although OBRS does not enforce that the actor distribution be identical to the policy distribution, it provably reduces the distance between the two distributions. It guarantees that for any rejection ratio, the adjusted actor distribution $q^*$ is strictly closer to the policy distribution $p$ than the unadjusted one $q$.

---

> ### Author Response · Authors · 2025-11-21
> **Official Comment to ZoQc (Part 2)**
>
> ## Q3 Contribution of each component
> We conduct ablation on masking and reweighting across a quantized actor model and large batch settings. Across both experiments, masking alone gives a decent boost over the vanilla baseline, but adding reweighting consistently pushes performance higher, especially when the setting suffers from severe staleness (e.g., large batch).
>
> **Experiment Setups**
> *(Model: Qwen3-4B-Base, C1 = C2 = 3.0, threshold c = 1.0, response limit = 8k, mini-batch/train-batch = 64/64, fp8 kv-quant, 40k examples)*
>
> | Experiments            | AIME24 | AMC    | MATH500 | GSM8K  |
> |------------------------|--------|--------|---------|--------|
> | van (best before crash) | 23.958 | 57.53  | 80.9    | 92.665 |
> | Masking-only           | 25.625 | 62.048 | 83.7    | 92.835 |
> | Masking & reweighting  | 26.667 | 62.651 | 82.45   | 92.305 |
>
> **Experiment Setups**
> *(Model: Qwen3-4B-Base, C1 = C2 = 3.0, threshold c = 1.0, response limit = 8k, mini-batch/train-batch = 64/2048, PPO clip = 0.4/0.7, 100k examples)*
>
> | Experiments                   | AIME24 |   AMC   | MATH500 | GSM8K  |
> |-------------------------------|--------|---------|---------|--------|
> | Masking-only (best before crash) | 19.167 | 49.699  | 78.75   | 91.793 |
> | Masking & reweighting         | 25.625 | 63.855  | 83.8    | 92.4   |
>
> ## Q4 Hyper-parameter choice
>
> Thanks for raising the question. We also included a discussion of hyperparameters in our modified paper $\\color{blue}{\\text{(Sections C.2 and C.3, p.g. 17-19)}}$.
>
> Our method involves three hyperparameters: $C_1$, $C_2$, and the rejection threshold $c$. All of them are straightforward to set, and the technique is robust across a wide range of choices.
>
> - **$C_1$**. We follow the standard truncated importance sampling choices. Empirically, we find that selecting $C_1$ in the range $[2, 10]$ consistently works well, and the method is **not sensitive** within this interval.
>               - **Explanation**: The parameter $C_1$ regulates the upper bound on the ratio $\frac{p_{\theta_{\mathrm{new}}}}{p_{\theta_{\mathrm{inf}}}}$ preventing situations in which $p_{\theta_{\mathrm{inf}}}$ is extremely small while $p_{\theta_{\mathrm{new}}}$ is not, which would otherwise cause this ratio to become unexpectedly large. In practice, however, such cases are **very rare** because rollouts are sampled directly from $p_{\theta_{\mathrm{inf}}}$, making tokens with extremely small $p_{\theta_{\mathrm{inf}}}$ unlikely to appear. Our empirical results also indicate that almost no tokens (**<5e-4**) are clipped by $C_1$.
>
> Experiment for $C_1$.
> （Model: Qwen3-4B-Base, C2=3.0, threshold c=1.0, response limit: 8k, mini batch/train batch: 64/2048, ppo clip: 0.4/0.7, 100k examples）
>
> - **$C_2$** (upper bound for $p_{\theta_{\mathrm{ref}}} / p_{\theta_{\mathrm{new}}}$). This parameter has **no practical effect** on performance. We set $C_2$ slightly larger than $1 + \varepsilon_{\mathrm{high}}$ (e.g., $1.28$ for DAPO), where any ratio clipped by $C_2$ would already be clipped by the trust region used in PPO. Thus, $C_2$ mainly serves as a **conceptual safeguard for stability**.
>
> - **Rejection threshold $c$**. Our method works well across all experiments with a default value of ($c = 1.0$), and we recommend choosing $c$ close to 1.0.
>                   - **Explanation**:  Increasing c makes the kept-token distribution closer to the target policy, but also increases the rejection rate. If the threshold is set above $1$, it will start rejecting tokens even when the policy and inference distributions are perfectly aligned, leading to overly conservative updates. The default setting ($c = 1.0$) guarantees full acceptance in the matched-distribution case and, in our experiments, already reduces the KL substantially while maintaining a high acceptance rate.
>
> **Experiment Setups**
> *(Model: Qwen3-4B-Base, C2 = 3.0, threshold c = 1.0, response limit = 8k, mini-batch/train-batch = 64/2048, PPO clip = 0.4/0.7, 100k examples)*
>
> ### Hyper-parameter Sweep (C1)
>
> | C1  | AIME24 |   AMC   | MATH500 | GSM8K  |
> |-----|--------|---------|---------|--------|
> | 2   | 26.875 | 63.855  | 82.8    | 92.267 |
> | 3   | 25.625 | 63.855  | 83.8    | 92.703 |
> | 4   | 26.042 | 65.06   | 83.5    | 92.684 |
> | 8   | 26.875 | 63.253  | 83.1    | 92.437 |
>
>
> ---
>
> **Experiment Setups**
> *Model: Qwen3-4B-Base (target) / Qwen3-1.7B-Base (rollout)*
> *Generation length limit: 8k*
> *Total training examples: 9k*
>
> ### Threshold Sweep
>
> | Threshold | AIME24 |  AMC  | MATH500 | GSM8K |
> |-----------|--------|-------|---------|-------|
> | 0.8       | 14.7   | 49.4  | 74.5    | 92.0  |
> | 0.9       | 12.5   | 47.0  | 74.6    | 91.8  |
> | 1.0       | 14.7   | 48.5  | 74.6    | 92.2  |
> | 1.1       | 12.3   | 47.4  | 74.1    | 92.0  |
> | 1.2       | 13.5   | 45.8  | 74.1    | 91.9  |

---

> > ### Author Response · Authors · 2025-11-21
> > **Official Comment to ZoQc (part 3)**
> >
> > ## Q5 Does the Phase 1 Data Collection with OBRS require modifications to the sampling code in vLLM or other inference frameworks?
> >
> > No, we directly use standard vLLM. vLLM provides an API to obtain top-k logits, which is the only extra information from vLLM we need beyond standard RL methods. https://docs.vllm.ai/en/v0.8.2/api/inference_params.html
> >
> > ## Q6 The LaTeX format of this paper does not follow the standard ICLR submission template.
> > Thank you for pointing this out. Our paper is prepared using the official ICLR LaTeX template and follows the required formatting guidelines. In response to the reviewer’s suggestion, we have further reviewed the manuscript and its layout settings to improve clarity and readability. We will ensure that the final version strictly adheres to the standard ICLR format.

---

### Author Response · Authors · 2025-11-21
**Response to all reviewers**

We thank all the reviewers [R1 (`ZoQc`)], [R2 (`BCWN`)], [R3 (`43dn`)], [R4 (`k9JV`)].  For their attention to our paper and their highly thoughtful and supportive feedback! We were glad that the reviewers found the work principled [`R1`, `R3`], with empirical promising results [`R1`, `R2`], theoretical guarantees [`R3`], tackling important and concrete problems [`R1`, `R3`, `R4`], and felt the findings are interesting [`R2`]. We are also glad to see that some reviewers find our presentation and narrative clean [`R4`].

In order to make sure that we take the reviewers’ comments seriously, we rewrote and modified our paper according to their feedback. We have updated the paper to incorporate constructive suggestions, as shown in the revision. We summarize the major changes:

**Hyper-parameter Robustness and Selection [`R1`, `R2`, `R3`, `R4`].**

Jackpot involves several important threshold and hyperparameter selections. We provide a detailed discussion of their role, selection, and sensitivity of the hyperparameters. We conduct comprehensive and large-scale hyperparameter ablation studies, demonstrating that our method is robust across a wide range of settings and generally insensitive to the selection of various hyperparameters. Additionally, we offer concrete recommendations for how to choose them in practice according to different user settings. $\\color{blue}{\\text{(Appendix C, p.g. 16-19)}}$

**Implementation and Additional Computation Analysis [`R1`, `R2`, `R3`].**

Our method can be seamlessly integrated into existing RL training pipelines and vLLM with minimal engineering effort. Additionally, our method DONOT require additional computation of log_probs and model inferences compared to standard PPO, since we show that we are reusing standard PPO computation outputs directly. In practice, the additional computation introduced by our approach amounts to only about 3% of a standard RL step, making the overhead negligible. Moreover, the total training time can be further reduced by leveraging large-batch training and performing rollouts with more efficient models, both of which significantly accelerate the overall workflow while maintaining performance.  $\\color{blue}{\\text{(Section 4.5, p.g. 7)}}$

**Motivation and novelty of applying OBRS in RL [`R1`, `R2`].**

We discussed the motivation for using OBRS in Jackpot to reduce the distribution mismatch that arises in RL training.  Moreover, previous TIS-based methods suffer from a large distribution gap between actor and policy mismatch, as the importance ratio forces the trajectories sampled by the actor of low likelihood, causing a training-inference mismatch. Thus, the issue motivates our method that directly adjusts and modifies the actor’s sampling tokens. Additionally, we also clarified that our use of OBRS is complementary to prior methods, rather than a replacement, of importance-sampling–based corrections (e.g., TIS). $\\color{blue}{\\text{(Section 1, p.g. 1, Section 2.2, p.g. 3)}}$

---

### Author Response · Authors · 2025-12-03
**A Call for AC's Discretion (Thanks to all the reviewer for your attention on Jackpot)**

We would like to express our wholehearted gratitude to all the reviewers who participated in the Jackpot review and to ACs for your additional effort in moderating and attention to our work. **Unfortunately, through the lengthy three weeks of rebuttal, we carefully drafted a response, but only one reviewer responded seconds before the review portal closed, while others remained silent and unresponsive.** Additionally, we are aware of the ongoing AC reassignment due to the cyberattack on OpenReview. To make our new AC’s life easier in navigating our lengthy paper and rebuttal history, here we summarize and organize key clarification points and additional experiments we added between our conversations with our reviewers. Four main concerns.

1. **Lack of analysis of the sensitivity and effectiveness of each of the hyperparameters, and the influence of each stage in the Jackpot** [`ZoQc`, `BCWN`, `43dn`, `k9JV`]

We conduct **extensive and large-scale experiments on the ablations of the different hyperparameters** used in our method, Jackpot, and we conclude from presenting our results that **the hyperparameters are, in general, insensitive to user selection and only there to fend off extremely rare scenarios**. In particular, hyperparameters of interest are C1, C2 inside the truncated important ratios, and threshold lambda inside the rejection sampling expression. Additionally, we also provide recommendations for them that can generally work well across diverse settings of actor-policy mismatch.

On the other hand, we also conducted a comprehensive ablation on the effectiveness of different stages of Jackpot, **rejection-only (without reweighting)**, and **rejection-plus-reweighting**. The ablations are also conducted across multiple settings of actor-policy misalignments. Through our experiments, we consistently demonstrated that **rejection-only (without reweighting) is itself more effective than the baseline setting without rejection, but still prone to early training instability and collapse. Both stages are indeed needed.**

2. **Concern about additional computation cost** [`ZoQc`, `BCWN`, `43dn`]

**We show both theoretically and empirically that our method can be seamlessly integrated into existing frameworks of vLLM and verl RL engines.** 1. Contrary to reviewers’ statements, Jackpot does NOT require any additional log prob computation (to `43dn`). We demonstrate that first, all the required log prob are already computed from the standard PPO computations and are reused by the Jackpot.  2. **Jackpot does NOT require additional trajectories to be sampled** (to `BCWN`). Our rejection is happening after generating the entire sequences (with the same number of trajectories as standard GRPO settings). The rejection decision is on a per-token basis and does not need any resampling. **Therefore, Jackpot incurs minimal overhead and NO additional probability or trajectory computations.** 3. Additionally, **we empirically validate the claim by benchmarking the per-step latency**; with less than 5% overhead, Jackpot brings significantly better training stability, convergence, and less training time (enabled by using cheaper rollout models).

3. **Motivation and novelty of OBRS to previous methods TIS** [`ZoQc`, `BCWN`]

We clarified that our use of OBRS is complementary to prior methods, rather than a replacement, of importance–sampling–based corrections (e.g., TIS).

**Previous TIS-based methods, although popular, aren’t very effective when the distribution gap between the actor and policy is large.** $\\color{blue}{\\text{(Figure 1, page 2)}}$ TIS suffers from a large distribution gap between actor and policy mismatch, as the importance ratio forces the trajectories sampled by the actor of low likelihood, causing a training-inference mismatch. **Thus, the issue motivates our method, OBRS, which directly adjusts and modifies the actor’s sampling tokens.** Empirically, we do show that OBRS indeed helps significantly alleviate the severe performance degradation introduced by TIS.

4. **Lack of illustration of the methods**

We are aware that some concerns raised by the reviewers are caused by a misunderstanding of the method presentation in the paper. To make sure reviewers understand the paper better, we invested significant effort in revising the paper: paragraph rewriting and adding clarifying figures.

We wish that the overview we presented is concise for reviewers and the AC. Our work, Jackpot, significantly improves the probability distribution alignment upon prior methods, even in extreme actor-policy mismatch settings. We believe that Jackpot will unlock RL training efficiency savings that were previously infeasible: using a completely different, smaller model’s rollout to train a much larger, more expensive model.
The value of Jackpot will be attended to by both the RL and efficiency communities. Jackpot will be the first step towards complete disentanglement between actor and policy models.

---

### Meta-Review · Area_Chair_Tz4d · 2026-01-07

**Summary:**

the reviewers generally found the problem important, the approach principled, and the empirical results strong, especially in extreme off-policy settings. key concerns revolved around the novelty compared to tis, the computational overhead, the sensitivity and role of hyperparameters (c1, c2, λ), and a desire for more analysis of the bias introduced by the practical stabilization heuristics. the authors provided extensive rebuttals, including new ablations and clarifications.

**Reviewer Concerns:**

the conceptual concerns from k9jv about clipping bias and visitation skew were partially addressed with data but remain as interesting discussion points. the lack of evaluation on non-mathematical tasks (raised by 43dn) is a remaining weakness.

**Reviewer Scores:**

zoqc: initial score 4. with concerns largely addressed, likely would raise to 5.

bcwn: initial score 4. after detailed overhead clarification, likely would raise to 5.

43dn: initial score 4. while compute concerns addressed, the bias analysis and lack of broader eval might keep score around 4.

k9jv: initial score 6. the reviewer's deeper methodological concerns were engaged but not fully resolved; score might stay around 6 or slightly lower to 5.

---

### Decision · Program_Chairs · 2026-01-26

Accept (Poster)